# Metropolis-Hastings Sampling for 3D Gaussian Reconstruction

**Hyunjin Kim**[1]     **Haebeom Jung**[2]     **Jaesik Park**[2]

[1]UC San Diego     [2]Seoul National University

hyk071@ucsd.edu     {haebeom.jung, jaesik.park}@snu.ac.kr

## Abstract

We propose an adaptive sampling framework for 3D Gaussian Splatting (3DGS) that leverages comprehensive multi-view photometric error signals within a unified Metropolis-Hastings approach. Vanilla 3DGS heavily relies on heuristic-based density-control mechanisms (e.g., cloning, splitting, and pruning), which can lead to redundant computations or premature removal of beneficial Gaussians. Our framework overcomes these limitations by reformulating densification and pruning as a probabilistic sampling process, dynamically inserting and relocating Gaussians based on aggregated multi-view errors and opacity scores. Guided by Bayesian acceptance tests derived from these error-based importance scores, our method substantially reduces reliance on heuristics, offers greater flexibility, and adaptively infers Gaussian distributions without requiring predefined scene complexity. Experiments on benchmark datasets, including Mip-NeRF360, Tanks and Temples and Deep Blending, show that our approach reduces the number of Gaussians needed, achieving faster convergence while matching or modestly surpassing the view-synthesis quality of state-of-the-art models. Our project page is available at https://hjhyunjinkim.github.io/MH-3DGS.

## 1   Introduction

Novel view synthesis (NVS) focuses on creating photo-realistic images of a 3D scene from unseen viewpoints. It is significant due to its broad applicability across various real-world scenarios. While Neural Radiance Fields (NeRF) [36] has largely advanced Novel View Synthesis by mapping 3D locations and view directions to view-dependent colors and volumetric densities, its slow rendering speed hinders real-world deployment. To address these limitations, 3D Gaussian Splatting (3DGS) [20] recently emerged, enabling real-time photo-realistic rendering by representing complex scenes with explicit 3D Gaussians, each of which is projected to the screen and defined by its position, anisotropic covariance, and opacity.

Despite these advantages, 3DGS and its extensions [18, 20, 56] rely heavily on heuristic-based adaptive density control mechanisms, such as cloning, splitting, and pruning by introducing fixed thresholds on opacity, size, and positional gradients in the view space. This leads to inflation of memory with redundant Gaussians when thresholds are loose, or to a sacrifice of fidelity when they are tight. To tackle this problem, Kheradmand et al. [22] recast 3DGS updates as Stochastic Gradient Langevin Dynamics (SGLD), framing Gaussian updates as Markov Chain Monte Carlo (MCMC) samples from an underlying probability distribution representing scene fidelity. As SGLD updates Gaussians in the direction of the image-reconstruction gradient and adds a small noise term, the procedure behaves like a gradient-based optimizer. Although it replaces heuristics with state transitions, it still fixes the total number of Gaussians upfront, limiting adaptability to scenes of varying complexity.

39th Conference on Neural Information Processing Systems (NeurIPS 2025).

Motivated by these considerations, we propose an adaptive sampling framework that uniquely leverages comprehensive multi-view error signals within a unified Metropolis-Hastings (MH) [6, 35] approach. We reformulate 3D Gaussian Splatting (3DGS) as an MH sampling process and newly derive a concise, rigorously proven acceptance probability grounded in the classical MH rule. By casting the densification and pruning stages of 3DGS as a probabilistic sampling problem, our method selectively adds Gaussians in regions with high visual or geometric significance as indicated by aggregated view-consistent photometric discrepancies. By deriving Gaussian-level importance scores from opacity and multi-view error metrics, we generate Gaussian candidates through coarse-to-fine proposals, which are subsequently accepted or rejected via Bayesian acceptance tests. This probabilistic process adaptively converges to a well-balanced Gaussian distribution with only minimal reliance on heuristics. It requires no prior knowledge of scene complexity while seamlessly integrating with existing 3DGS workflows to achieve better reconstruction quality, capturing both coarse structures and fine details.

We demonstrate the effectiveness of our approach on real-world scenes from the Mip-NeRF360 [1], Tanks and Temples [23], and Deep Blending [17] datasets. Our method reduces the number of Gaussians needed to represent each scene solely through an improved sampling mechanism, while consistently matching or surpassing the view-synthesis quality of 3DGS across diverse scenes. Additionally, our approach converges faster than 3DGS-MCMC [22], achieving target PSNR values in fewer training iterations and reduced wall-clock time. By coupling principled probabilistic inference with dense multi-view error signals, our method matches the rendering accuracy of 3DGS-MCMC [22] with fewer Gaussians, demonstrating that comparable quality can be achieved with a leaner, more generalizable representation that converges more rapidly.

We summarize our contributions as follows:

- We introduce an adaptive Metropolis–Hastings (MH) framework that replaces densification heuristics in 3DGS with a closed-form Bayesian posterior for each Gaussian and a lightweight photometric surrogate that turns the MH acceptance ratio into a single logistic–voxel product.

- To this end, we derive Metropolis–Hastings transition equations that are specialized to 3DGS and prove that this MH sampler is mathematically sound, demonstrating that the heuristic densification rule in 3DGS can be recast as a principled MH update.

- Our experiments show that our framework achieves compelling view synthesis quality while converging faster.

## 2  Related Work

### 2.1  Neural 3D Scene Representation

Novel View Synthesis generates realistic images from novel viewpoints that differ from the original captures [56]. Neural Radiance Fields (NeRF) [36] pioneered this field by using MLPs to model 3D scenes from multi-view 2D images, inspiring extensions for large scenes [13, 46, 52], dynamic content [8, 14, 40, 41, 49], and pose-free reconstruction [2, 19, 27, 33, 34, 57]. However, NeRF's extensive training and rendering time prompted numerous acceleration efforts [4, 10, 11, 37, 42, 45].

3D Gaussian Splatting (3DGS) [20] emerged as a compelling alternative, using differentiable rasterization of 3D Gaussian primitives initialized from sparse point clouds generated from Structure-from-Motion [44]. This approach enables efficient optimization and high-resolution output, spurring extensions in dynamic scenes [25, 26, 31, 51, 53], 3D generation [5, 29, 47, 48, 55], and large-scale reconstruction [21, 28, 30]. Yet, 3DGS suffers from a significant memory burden due to covariance matrices and spherical harmonics, leading to memory-efficient variants that rely on heuristic density-control rules [9, 15, 16, 24, 38, 39, 50].

Recent work has explored probabilistic sampling in 3D reconstruction: Goli et al. [12] and Bortolon et al. [3] apply Metropolis-Hastings for sample refinement and surface point selection in NeRF. While these methods use Metropolis-Hastings for specific components, our work reformulates the entire 3D reconstruction problem as a Metropolis-Hastings sampling process, replacing heuristic density controls in 3DGS with a unified probabilistic framework that enhances both efficiency and generality.

## 2.2 Adaptive Density Control for 3D Gaussian Splatting

Kheradmand et al. [22] recently introduced the use of Stochastic Gradient Langevin Dynamics (SGLD) for sampling and optimization in 3DGS, combined with an opacity-driven relocation strategy to shift low-opacity Gaussians towards higher-opacity regions, while enforcing a fixed limit on the total number of Gaussians. Additionally, Rota Bulò et al. [43] introduces error-based densification with perceptual error prioritization and sets a global limit on the number of Gaussians. Similarly, Ye et al. [54] addresses the limitations of gradient collision in adaptive density control by proposing a homodirectional gradient as a criterion for densification. Moreover, Mallick et al. [32] presents a score-based guided densification approach that uses training-time priors to restrict growth. More recently, Deng et al. [7] introduced deterministic geometric splitting rules for efficient density control. As 3DGS-MCMC [22] aligns most closely with our objectives and delivers the best overall reconstruction quality, it serves as our primary baseline.

In contrast, our proposed framework leverages a unified Metropolis-Hastings sampling strategy grounded in comprehensive multi-view error signals. Rather than relying on thresholds, our method computes Gaussian importance scores from aggregated multi-view photometric errors and dynamically adapts the number and distribution of Gaussians via probabilistic inference. We derive the MH acceptance rule in the context of 3DGS and prove that our importance-weighted proposals preserve detailed balance with respect to the multi-view photometric likelihood. This rigorous formulation distinguishes our framework from prior methods and enables natural scaling with scene complexity.

# 3 Preliminary

## 3.1 3D Gaussian Splatting

3D Gaussian Splatting (3DGS) [20] represents the 3D scene as a set of anisotropic 3D Gaussians, further optimized by differentiable tile rasterization. Each 3D Gaussian $\mathcal{G}$ is characterized by a position vector $\mu$ and a covariance matrix $\Sigma$:

$$\mathcal{G}(x) = e^{-\frac{1}{2}(x-\mu)^\top \Sigma^{-1}(x-\mu)}, \quad \Sigma = RSS^T R^T, \tag{1}$$

where the covariance matrix is factorized into a scaling matrix $S$ and a rotation matrix $R$. The transformed 2D covariance matrix $\Sigma'$ is calculated by the viewing transform $W$ and the Jacobian of the affine transformation of the projective transformation $J$ as $\Sigma' = JW\Sigma W^T J^T$. The color $C$ of a pixel can be computed by blending $\mathcal{N}$ ordered points overlapping the pixel as $C = \sum_{i \in \mathcal{N}} c_i \alpha_i \prod_{j=1}^{i-1}(1 - \alpha_j)$, where $c_i$ and $\alpha_i$ represent the view-dependent color and opacity computed from a 2D Gaussian distribution.

To achieve a high-quality representation, 3DGS employs adaptive density control that clones Gaussians in under-reconstructed areas and splits Gaussians in over-reconstructed regions. During densification, 3DGS evaluates the positional gradient $\nabla_p \mathcal{L}$ of each Gaussian, refining further if the gradient exceeds a predefined threshold $\tau_p$. If a Gaussian surpasses the size threshold $\tau_s$, indicating over-reconstruction, it is split into smaller Gaussians. Alternatively, under-reconstructed areas prompt cloning by shifting Gaussians along the gradient direction for better scene representation.

## 3.2 Metropolis-Hastings Algorithm

The Metropolis-Hastings algorithm [6, 35] is a widely used Markov Chain Monte Carlo (MCMC) method for sampling from complex, high-dimensional probability distributions. The objective is to construct a Markov chain whose stationary distribution $\pi(\mathbf{x})$ matches the target distribution, enabling efficient sampling where direct methods are impractical.

Given a target distribution $\pi(\mathbf{x})$ defined over a state space $\mathcal{X}$, the algorithm generates a sequence $\{\mathbf{x}^{(t)}\}_{t=1}^N$ of samples by iteratively proposing candidates from a proposal distribution $q(\mathbf{x}, \mathbf{y})$ and accepting or rejecting them based on an acceptance probability. At each iteration $t$, a new candidate $\mathbf{y}$ is proposed from the distribution $q(\mathbf{x}^{(t)}, \cdot)$. The acceptance probability is further computed as

$$\rho(\mathbf{x}^{(t)}, \mathbf{y}) = \min\left(1, \frac{\pi(\mathbf{y})q(\mathbf{y}, \mathbf{x}^{(t)})}{\pi(\mathbf{x}^{(t)})q(\mathbf{x}^{(t)}, \mathbf{y})}\right). \tag{2}$$

The candidate $\mathbf{y}$ is accepted with probability $\rho(\mathbf{x}^{(t)}, \mathbf{y})$; otherwise, the chain remains at the current state, $\mathbf{x}^{(t+1)} = \mathbf{x}^{(t)}$. The acceptance probability is designed to ensure that the Markov chain satisfies the detailed balance condition:

$$\pi(\mathbf{x})q(\mathbf{x}, \mathbf{y})\rho(\mathbf{x}, \mathbf{y}) = \pi(\mathbf{y})q(\mathbf{y}, \mathbf{x})\rho(\mathbf{y}, \mathbf{x}) \tag{3}$$

which guarantees that $\pi(\mathbf{x})$ is the stationary distribution of the Markov chain. The distribution of the generated samples converges to the target distribution $\pi(\mathbf{x})$ as the number of iterations $N$ increases.

## 4    Method

Our method begins with a key design choice: unlike 3DGS, which densifies points at fixed thresholds, we employ a lean point-based sampler that keeps the most informative Gaussians, sharply reducing storage while preserving fidelity. Section 4.1 lays the theoretical groundwork for this: we extend the Metropolis–Hastings algorithm to 3DGS and rigorously derive scene-adaptive equations that generalize the traditional formulation. Section 4.2 describes how the sampler proposes candidates guided by multi–view photometric and opacity errors, and Sec. 4.3 presents the MH acceptance test with a newly derived practical expression for the MH acceptance probability. The complete algorithmic flow is in Algorithm 1 and 2 in Appendix D.

### 4.1    Formulation of the Metropolis-Hastings Framework

Our main objective is to eliminate heuristic dependence in adaptive density control by recasting densification and pruning as a principled sampling problem. Specifically, we interpret each modification of the Gaussian set as a proposal in a Metropolis-Hastings Markov Chain. Each step in the chain perturbs the current scene representation $\Theta$ by *(i)* **inserting** new Gaussians near pixels that still exhibit high multi–view error or *(ii)* **relocating** Gaussians whose opacity has collapsed. The proposed scene $\Theta'$ is then accepted or rejected according to a probabilistic rule that, in expectation, favors configurations that reproduce the captured images better while remaining spatially sparse.

In our formulation, a scene is described as a collection $\Theta = \{g_i\}_{i=1}^{N}$ of Gaussian splats, where each splat contains parameters $(\mathbf{x}_i, \Sigma_i, \mathbf{c}_i, \alpha_i)$, representing the position, covariance, color, and opacity of a Gaussian. We define the overall loss function as:

$$\mathcal{L}(\Theta) = (1 - \lambda)\mathcal{L}_1 + \lambda\mathcal{L}_{\text{D-SSIM}} + \lambda_{\text{opacity}}\overline{\alpha} + \lambda_{\text{scale}}\overline{\Sigma} \tag{4}$$

where $\mathcal{L}_1$ and $\mathcal{L}_{\text{D-SSIM}}$ are the loss terms from the original 3DGS [20] method. The last two terms are regularizers on the mean opacity and covariance, used to penalize large or widespread opacities and large Gaussian spreads, respectively, as advocated by 3DGS-MCMC [22]. We use $\lambda = 0.2$, $\lambda_{\text{opacity}} = 0.01$ and $\lambda_{\text{scale}} = 0.01$ in our framework.

From a likelihood viewpoint, we treat the minimization of $\mathcal{L}(\Theta)$ equivalent to maximizing the likelihood of the observed image set $\mathcal{D} = \{I^{(v)}\}$. Thus, we can interpret this as:

$$\ln p(\mathcal{D} \mid \Theta) \approx -\mathcal{L}(\Theta) \tag{5}$$

Since likelihood terms are usually expressed with base $e$, we adopt the natural logarithm ($\ln$) throughout to keep the formulas clean and consistent.

Furthermore, photometric fidelity alone does not stop the optimizer from piling Gaussians onto the same patch once it converges locally. Thus, we discourage such redundancies by introducing a voxel-wise prior:

$$\ln p(\Theta) = -\lambda_v \sum_{v \in \mathcal{V}} \ln(1 + c_\Theta(v)) \tag{6}$$

where $c_\Theta(v)$ is the number of Gaussians in voxel $v$. This prior encourages at most one or two Gaussians per voxel. Since the cost grows logarithmically, empty cells get almost no penalty, but the penalty rises quickly once the voxel becomes crowded. Combining Eq. 4 and 6, we are able to define a negative log-posterior:

$$\mathcal{E}(\Theta) = \mathcal{L}(\Theta) + \lambda_v \sum_{v \in \mathcal{V}} \ln(1 + c_\Theta(v)) \tag{7}$$

Based on this derivation, we can define the posterior density based on Bayesian statistics:

$$\pi(\Theta) = \frac{1}{Z} e^{-\mathcal{E}(\Theta)}, \tag{8}$$

where $Z$ is a normalization constant (See Appendix A for details). Our framework samples Gaussian configurations $\Theta$ in proportion to $\pi(\Theta)$ and adaptively refines the 3D Gaussian representation to cover undersampled regions. To facilitate this sampling, we construct a Markov Chain whose states are successive scene representations $\Theta_0, \Theta_1, \ldots, \Theta_t$. From a given state $\Theta_t$, we draw a proposal $\Theta'$ by adding Gaussians in high-error areas or relocating low opacity Gaussians, with probability density $q(\Theta' \mid \Theta_t)$. The construction of $q(\Theta' \mid \Theta_t)$ is detailed in Sec. 4.2.

We accept or reject $\Theta'$ according to the Metropolis-Hastings (MH) rule:

$$\rho(\Theta' \mid \Theta_t) = \min\left(1, \frac{\pi(\Theta')\, q(\Theta_t \mid \Theta')}{\pi(\Theta_t)\, q(\Theta' \mid \Theta_t)}\right), \quad \rho(\Theta' \mid \Theta_t) \in [0, 1] \tag{9}$$

where $\rho(\Theta' \mid \Theta_t)$ indicates the standard MH acceptance probability. At each step we draw $u \sim \mathcal{U}(0, 1)$; if $u < \rho$, the proposal is accepted and we set $\Theta_{t+1} = \Theta'$, otherwise we keep $\Theta_{t+1} = \Theta_t$. We explain how the MH rule accepts or rejects $\Theta'$ again in Sec. 4.3. Over many iterations, this Markov Chain converges toward our target distribution $\pi(\Theta)$, systematically densifying the Gaussian set, reducing loss, and relocating Gaussians that fail to improve image fidelity.

## 4.2 Proposal Generation

Building on the Metropolis-Hastings framework, we now specify how a candidate configuration $\Theta'$ is generated from the current state $\Theta$. The guiding principle is to concentrate proposals in problematic regions where $\Theta$ either (1) lacks sufficient **opacity**, indicating missing or overly sparse geometry, or (2) exhibits a large **photometric error**.

### 4.2.1 Per-pixel importance field

At every densification step, we first choose a working view subset $\mathcal{C}_t = \{c_{t,1}, \ldots, c_{t,k_t}\}$ from the full training set $\mathcal{C}$ to ensure balanced use of all viewpoints over time and broad viewing-angle coverage. Note that this selected subset $\mathcal{C}_t$ is used only for computing the importance field. The subset size is annealed according to

$$k_t = \max\left(1, \lfloor (1 - \eta_t) |\mathcal{C}| \rfloor\right), \qquad \eta_t = \frac{t - t_{\min}}{t_{\max} - t_{\min}} \tag{10}$$

so that early iterations ($\eta_t \approx 0$) ensure broad coverage across diverse views, whereas we reduce the subset size to focus on specific vantage points requiring finer coverage. We use a *round-robin* schedule that walks through $\mathcal{C}$ in contiguous blocks of length $k_t$, as this allows every viewpoint to contribute equally to the error signal after a few iterations. For each viewpoint $c \in \mathcal{C}_t$, we compare the current prediction $\hat{I}^{(c)}$ to the ground truth image $I^{(c)}$ and record per-pixel SSIM and $\mathcal{L}_1$ errors. To construct a unified importance field for Gaussian proposal generation, we aggregate these error maps and construct view-averaged importance maps:

$$\text{SSIM}_{\text{agg}}(p) = \frac{1}{k_t} \sum_{c \in \mathcal{C}_t} \left[1 - \text{SSIM}(I^{(c)}(p), \hat{I}^{(c)}(p))\right], \quad \mathcal{L}1_{\text{agg}}(p) = \frac{1}{k_t} \sum_{c \in \mathcal{C}_t} \left|I^{(c)}(p) - \hat{I}^{(c)}(p)\right| \tag{11}$$

where $k$ is the number of selected viewpoints, and $p$ indicated image pixels. This aggregation produces a spatial field that captures the difficulty of reconstruction across the current viewing subset. We then feed $\text{SSIM}_{\text{agg}}(p)$ and $\mathcal{L}1_{\text{agg}}(p)$ along with the opacity $O(p)$, which is directly obtained by projecting the per-Gaussian opacity on pixel $p$, into our importance score. After applying robust normalization to each quantity (to mitigate outliers and match dynamic ranges), the three cues are fused through a logistic function:

$$s(p) = \sigma\left(\alpha\, O(p) + \beta\, \text{SSIM}_{\text{agg}}(p) + \gamma\, \text{L1}_{\text{agg}}(p)\right), \qquad \sigma(z) = \frac{1}{1 + e^{-z}} \tag{12}$$

where $\alpha, \beta, \gamma$ control the relative emphasis on opacity, structural similarity, and photometric fidelity, respectively. We use $\alpha = 0.8, \beta = 0.5, \gamma = 0.5$ in our tests. The scalar field $s \in (0, 1)$ highlights the pixels that are simultaneously under-covered and photometrically inaccurate across the chosen views.

### 4.2.2 Mapping Pixels to Gaussian Importance

Given the 3D position $\mathbf{x}_i \in \mathbb{R}^3$ of a Gaussian $g_i$, we obtain its image-plane coordinates via the calibrated projection $\Pi \colon \mathbb{R}^3 \to \mathbb{R}^2$. Rather than integrating the importance field over the full elliptical footprint of $g_i$, which requires evaluating hundreds of pixels per splat, we adopt a lightweight surrogate that uses the floor operator to sample only the pixel lying directly beneath the center of $g_i$:

$$I(i) \;=\; s\big(\lfloor \Pi(\mathbf{x}_i) \rfloor\big) \tag{13}$$

where $s(\cdot)$ is the logistic importance map in Eq. 12. The importance weight $I(i)$ is then normalized once per iteration to give a categorical distribution $P_{\text{pick}}(i) = I(i)/\sum_j I(j)$.

### 4.2.3 Proposal Distribution

Having established a Bayesian viewpoint on our 3D Gaussian representation and a per-pixel importance score $s(p)$, we now detail the proposal mechanism in our Metropolis-Hastings (MH) sampler. We maintain a Markov Chain $\Theta_0 \to \Theta_1 \to \dots$, each $\Theta_t$ comprising the current Gaussian set. From state $\Theta_t$, we form a proposal $\Theta'$ in **two** phases.

During the **Coarse Phase**, we sample a batch $\mathcal{I}_c = \{i_1^c, \dots, i_{B_c}^c\}$ of size $B_c$ from $P_{\text{pick}}$. For every index $i \in \mathcal{I}_c$, we create a new Gaussian:

$$g_i'^c \;=\; \big(\mathbf{x}_i + \boldsymbol{\delta}_i^c, \Sigma_i, \mathbf{c}_i, \alpha_i\big), \quad \boldsymbol{\delta}_i^c \sim \mathcal{N}(\mathbf{0}, \sigma_{\text{coarse}}^2 \mathbf{I}) \tag{14}$$

where $\sigma_{\text{coarse}}$ is chosen to be large enough to enable the sampler to explore broad coverage gaps.

In the **Fine Phase**, a second batch $\mathcal{I}_f = \{i_1^f, \dots, i_{B_f}^f\}$ is also drawn from the distribution $P_{\text{pick}}$. Similar to the coarse phase, we create a new Gaussian for every index in $i \in \mathcal{I}_f$:

$$g_i'^f \;=\; \big(\mathbf{x}_i + \boldsymbol{\delta}_i^f, \Sigma_i, \mathbf{c}_i, \alpha_i\big), \quad \boldsymbol{\delta}_i^f \sim \mathcal{N}(\mathbf{0}, \sigma_{\text{fine}}^2 \mathbf{I}) \tag{15}$$

where $\sigma_{\text{fine}} < \sigma_{\text{coarse}}$. The smaller perturbation radius focuses on residual high-importance regions that still yield large weights $I(i)$ (and hence a large $P_{\text{pick}}(i)$) after the coarse phase fills the major coverage gaps. This allows our method to refine the geometry further using locally placed Gaussians. See Fig. 2 for a visualization of the coarse–fine proposal phase.

Finally, all the newly spawned Gaussians are combined to form the birth component $\Theta'_{\text{birth}}$ of the proposal. This is combined the current state $\Theta_t$ to yield a full proposal $\Theta'$:

$$\Theta'_{\text{birth}} = \big\{g_i'^c \,\big|\, i \in \mathcal{I}_c\big\} \;\cup\; \big\{g_i'^f \,\big|\, i \in \mathcal{I}_f\big\}, \quad \Theta' = \Theta_t \;\cup\; \Theta'_{\text{birth}} \tag{16}$$

Formally, the probability of drawing the birth component is:

$$q(\Theta' \mid \Theta) = \prod_{i \in \mathcal{I}_c \cup \mathcal{I}_f} P_{\text{pick}}(i)\, \mathcal{N}(\mathbf{x}_i'; \mathbf{x}_i, \sigma_\phi^2 \mathbf{I}) \tag{17}$$

### 4.2.4 Relocation of Low Contributing Gaussians

In addition to densification, our framework incorporates a relocation mechanism inspired by 3DGS-MCMC [22] to reuse Gaussians with persistently low opacity. Let $\mathcal{D} = \{i \mid \alpha_i \le \tau\}$ denote the set of Gaussians with low opacity, and $\mathcal{A} = \{j \mid \alpha_j > \tau\}$ the set of high opacity Gaussians. We consider $\tau = 0.005$. For each $g_i \in \mathcal{D}$, we sample a replacement index $j \in \mathcal{A}$ with probability

$$p(j) = \frac{\alpha_j}{\sum_{k \in \mathcal{A}} \alpha_k},$$

and update $g_i$'s parameters by transferring those from $g_j$, optionally modulated by a ratio reflecting the sampling frequency. The relocation step preserves the representational fidelity of our framework by continuously reinforcing effective Gaussians while reassigning under-contributing ones.

## 4.3 Metropolis-Hastings Proposal Acceptance

Now, the Metropolis-Hastings sampler must decide whether the proposal $\Theta'$ is kept or rejected. Recall the MH rule in Eq. 9. Considering the posterior density in Eq. 8, we can rewrite the rule as:

$$\rho_{\text{MH}} = \min\Big\{1,\, e^{-\Delta\mathcal{E}} \frac{q(\Theta \mid \Theta')}{q(\Theta' \mid \Theta)}\Big\} \tag{18}$$

Figure 1: PSNR over training iterations for our method and 3DGS-MCMC, averaged across all scenes. Dotted vertical lines indicate when each method reaches 98 % of its eventual PSNR, highlighting near-optimal convergence speed. While both methods ultimately attain comparable final PSNR, ours converges faster.

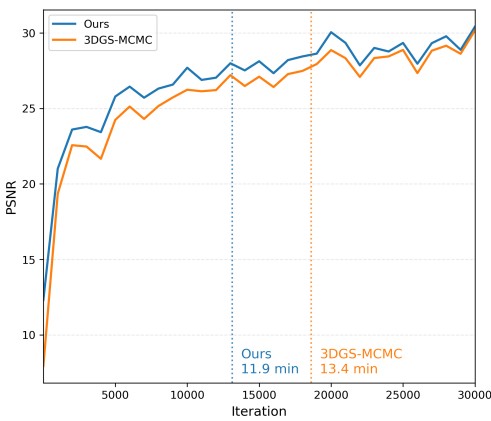

Table 1: Time to equal PSNR comparison between our method and 3DGS-MCMC, averaged across all benchmark scenes. Our method consistently reaches target PSNR thresholds faster.

| Target PSNR (dB) | Time - Ours (s / mins) | Time - 3DGS-MCMC (s / mins) |
|---|---|---|
| 21 | **16.30 / 0.27** | 17.08 / 0.28 |
| 24 | **61.34 / 1.02** | 98.38 / 1.64 |
| 27 | **287.01 / 4.78** | 341.64 / 5.69 |
| 30 | **851.52 / 14.19** | 983.05 / 16.38 |

Figure 2: 3D visualization of (a) coarse and (b) fine proposal stages (see Sec. 4.2.3 for details).

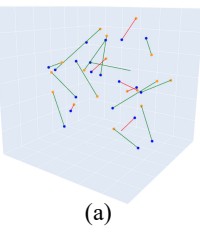 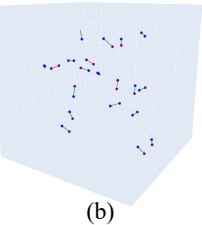

(a)           (b)

where $\Delta\mathcal{E} = \mathcal{E}(\Theta') - \mathcal{E}(\Theta)$. The proposal $\Theta'$ is a batch of new Gaussians, but since the perturbations are independent, we can evaluate the acceptance separately. Considering one candidate Gaussian whose center falls in voxel $v'$, the insertion of that particular Gaussian leads to $\Delta\mathcal{E}$ as:

$$\Delta\mathcal{E} = \Delta\mathcal{L} + \lambda_v \ln\left(\frac{1 + c_\Theta(v') + 1}{1 + c_\Theta(v')}\right) = \Delta\mathcal{L} - \ln D(v'), \quad D(v') = \frac{1}{1 + \lambda_v c_\Theta(v')} \quad (19)$$

where $\Delta\mathcal{L}$ is the photometric change and $c(v')$ is the voxel occupancy before the insertion. (See Appendix B for details.) However, evaluating $\mathcal{L}$ requires rendering every view in the set $\mathcal{C}_t$, which is computationally expensive. Thus, we use the tight empirical correlation between the importance map $I(i)$ and the loss reduction, $-\Delta\mathcal{L} \approx I(i)$, as a surrogate. (See Appendix C for details.)

Moreover, the calculation of $q(\Theta \mid \Theta')$ in Eq. 18 requires the enumeration of the reverse action, which is deleting the particular newborn Gaussian. As our framework does not generate such a move, we absorb the unavailable reverse-proposal density into the voxel factor $D(v') = (1 + \lambda_v c_\Theta(v'))^{-1}$, which already penalises insertions in crowded cells. Substituting both surrogates into Eq. 18 gives an upper bound $\min\{1, e^{I(i)} D(v')\}$, and mapping the exponential through the logistic $\sigma(z) = (1 + e^{-z})^{-1}$ yields a practical version of the MH rule:

$$\rho(i) = \sigma\big(I(i)\big) D(v') \quad (20)$$

Following the Metropolis-Hastings algorithm, we draw a uniform random number $u \sim \mathcal{U}(0, 1)$ for each candidate Gaussian. The proposal is accepted and added to the current set $\Theta_t$ when $u < \rho(i)$, otherwise it is rejected. Because $\rho(i) = \sigma(I(i)) D(v')$ is large only when the per-Gaussian importance $I(i)$ is high and the voxel factor $D(v')$ is close to one, Gaussians that target significant photometric errors in sparsely populated regions are accepted with high probability, whereas candidates in already crowded voxels or low-error areas are mostly discarded.

## 4.4 Comparison with 3DGS-MCMC [22]

Although both methods fall under the broad scope of Monte-Carlo sampling, they embody *fundamentally different* MCMC philosophies. 3DGS-MCMC treats every Gaussian parameter as a latent variable and applies SGLD: it perturbs all dimensions with small, isotropic noise and accepts every proposed move, so the chain drifts locally through the existing cloud.

In contrast, our method runs a Metropolis-Hastings birth sampler that targets only positions. We draw global proposals straight in image-space voids highlighted by a multi-view error map, then pass each

Table 2: Quantitative results of our method and state-of-the-art models evaluated on Mip-NeRF 360 [1], Tanks&Temples [23] and Deep Blending [17] datasets. We present the baseline results from 3DGS [20]. For a fair comparison, we also report the results for 3DGS obtained from our experiments (Denoted as 3DGS*), as well as those for 3DGS and 3DGS-MCMC trained with the *same number of Gaussians* as our method (denoted as 3DGS-S and 3DGS-MCMC, respectively). The unit of the number of Gaussians is million. The results are colored as best , second-best , and third-best . To avoid redundancy, only 3DGS* results are considered instead of 3DGS.

| Dataset | Mip-NeRF 360 | | | | Tanks & Temples | | | | Deep Blending | | | |
|---|---|---|---|---|---|---|---|---|---|---|---|---|
| Method | PSNR↑ | SSIM↑ | LPIPS↓ | # GS↓ | PSNR↑ | SSIM↑ | LPIPS↓ | # GS↓ | PSNR↑ | SSIM↑ | LPIPS↓ | # GS↓ |
| Plenoxels [10] | 23.08 | 0.626 | 0.463 | - | 21.08 | 0.719 | 0.379 | - | 23.06 | 0.795 | 0.510 | - |
| INGP-base [37] | 25.30 | 0.671 | 0.371 | - | 21.72 | 0.723 | 0.330 | - | 23.62 | 0.797 | 0.423 | - |
| INGP-big [37] | 25.59 | 0.699 | 0.331 | - | 21.92 | 0.745 | 0.305 | - | 24.96 | 0.817 | 0.390 | - |
| Mip-NeRF 360 [1] | 27.69 | 0.792 | 0.237 | - | 22.22 | 0.759 | 0.257 | - | 29.40 | 0.901 | 0.245 | - |
| 3DGS [20] | 27.21 | 0.815 | 0.214 | - | 23.14 | 0.841 | 0.183 | - | 29.41 | 0.903 | 0.243 | - |
| 3DGS* [20] | 27.44 | 0.811 | 0.223 | 3.176 | 23.63 | 0.848 | 0.177 | 1.831 | 29.53 | 0.904 | 0.244 | 2.815 |
| 3DGS-S [20] | 26.62 | 0.761 | 0.299 | 0.723 | 23.38 | 0.828 | 0.214 | 0.773 | 29.27 | 0.897 | 0.272 | 0.751 |
| 3DGS-MCMC [22] | 27.69 | 0.816 | 0.234 | 0.723 | 24.30 | 0.862 | 0.170 | 0.773 | 29.84 | 0.906 | 0.254 | 0.751 |
| Ours | 27.34 | 0.798 | 0.241 | 0.723 | 23.99 | 0.852 | 0.166 | 0.773 | 30.12 | 0.909 | 0.245 | 0.751 |

candidate through a Metropolis-Hastings accept–reject test that weighs its predicted photometric gain against a sparsity prior. This decouples exploration ("where to propose") from retention ("what to keep"), enabling long jumps to uncovered regions and the principled dismissal of redundant splats, which Langevin-noise walks with unconditional acceptance cannot achieve. Moreover, we analytically derive and rigorously prove that the 3DGS densification process can be recast as a Metropolis–Hastings sampler, establishing a principled bridge between 3DGS and our formulation.

In principle, 3DGS-MCMC is a local, always-accept SGLD chain, whereas our approach is a global, importance-driven MH chain focused on the scene's unexplored areas. This leads to our method converging faster, as shown in Fig. 1. Measuring time to equal PSNR across all benchmark scenes, our method reaches 30 dB approximately 2.2 minutes faster than 3DGS-MCMC, with consistently faster convergence at intermediate quality thresholds, as shown in Tab. 1.

## 5 Experiments

### 5.1 Experimental Settings

**Dataset and Metrics.** We perform comprehensive experiments using the same real-world datasets employed in 3DGS. Specifically, we utilize the scene-scale view synthesis dataset from Mip-NeRF360 [1], which includes nine large-scale real-world scenes comprising five outdoor and four indoor environments. Additionally, we selected two scenes each from the Tanks and Temples [23] and the Deep Blending [17] datasets, using the same scenes as in the original 3DGS. We evaluate each method using the standard evaluation metrics: peak signal-to-noise ratio (PSNR), structural similarity index measure (SSIM), and learned perceptual image patch similarity (LPIPS).

**Implementation Details.** Our framework is based on 3DGS [20] implementation. We also use the differentiable tile rasterizer implementation provided by 3DGS-MCMC [22]. We conducted experiments on an NVIDIA RTX 3090 GPU. Averaged across all datasets, our method runs at 0.0528 seconds per iteration and 26.4 minutes for 30K iterations. During densification, a normalized progress value ($\in [0, 1]$) linearly transitions parameters from coarse (larger offsets, voxel sizes, and batch sizes) to fine (smaller offsets and voxel sizes) to shift focus from broad to precise refinements. Coarse proposal scales range from 10.0 to 5.0, fine proposal scales from 2.0 to 1.0, and voxel sizes from 0.02 to 0.005. Batch sizes for coarse and fine proposals are set to 4,500 and 16,000, respectively.

**Comparison with Baselines.** We compare against methodologies that model large-scale scenes. We compare our approach against state-of-the-art baselines, such as Plenoxels [10], Instant-NGP [37], Mip-NeRF 360 [1], and 3DGS [20]. For the baselines aside from 3DGS, we present the original results as reported in 3DGS. Moreover, we include a comparison with 3DGS-MCMC [22], which has a similar motivation and shows state-of-the-art performance.

Figure 3: Qualitative results of our method compared to 3DGS and the corresponding ground truth image from held-out test views. As shown in the red bounding box, our method better captures the fine and coarse details that 3DGS [20] and 3DGS-MCMC [22] miss.

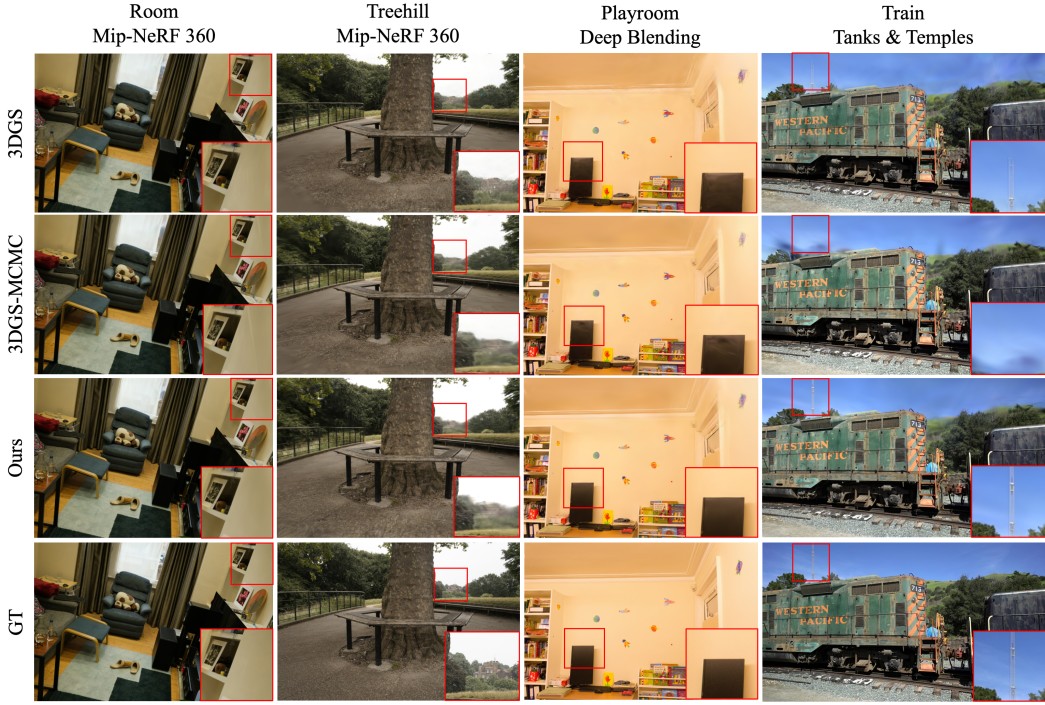

## 5.2 Results

Table 2 presents the quantitative evaluations, and Fig. 3 demonstrates the qualitative evaluations on real-world scenes. Our method consistently surpasses 3DGS performance with the same number of Gaussians and achieves or exceeds 3DGS performance with fewer points. Unlike 3DGS, which relies on fixed thresholds for densification, our framework leverages a streamlined point-based sampling strategy that drastically reduces storage requirements while preserving high-fidelity reconstructions. In particular, Tab. C in Appendix F demonstrates our approach's consistent efficiency across varied scene complexities. We also provide the full per-scene evaluations in Appendix E.

Although 3DGS-MCMC can yield slightly higher metrics, it updates and optimizes every Gaussian parameter, which increases its inefficiency and necessitates knowing the scene's complexity (the number of Gaussians) in advance. In contrast, our method focuses exclusively on sampling and optimizing point positions, providing a simpler yet equally accurate alternative. As evidenced by Fig. 1, our method reaches the 98 % PSNR threshold faster than 3DGS-MCMC, both in iteration and wall-clock time, demonstrating faster convergence. This shows that our framework can still achieve state-of-the-art reconstruction quality even without exhaustive parameter updates.

## 5.3 Ablation Study

**Components for the Importance Scores.** We conducted an ablation study on the key Gaussian components to identify the most influential factors in importance score calculation for accurate sampling. The results, shown in Tab. 3, support the effectiveness of our *Importance Score Computation* design. Specifically, combining SSIM, L1 loss, and opacity values yields higher PSNR scores and reduces the required number of Gaussians compared to both the baseline and variants that rely solely on photometric losses. This study underscores the need for careful factor selection.

**Effectiveness of Relocation.** We explore whether relocating Gaussians was effective in Tab. 5. Our experiments confirm that the relocation strategy is essential for maintaining high representational fidelity. It shows that dynamically reallocating underperforming Gaussians to areas where they contribute more effectively improves performance in both indoor and outdoor scenes.

Table 3: Ablations on the importance score components.

| Method | Dataset | | Mip-NeRF360 [1] | | | Tanks&Temples [23] | | | Deep Blending [17] | | |
|---|---|---|---|---|---|---|---|---|---|---|---|
| L1 | SSIM | Opacity | PSNR↑ | SSIM↑ | LPIPS↓ | PSNR↑ | SSIM↑ | LPIPS↓ | PSNR↑ | SSIM↑ | LPIPS↓ |
| ✓ | ✓ | | 27.22 | 0.796 | 0.243 | 23.89 | 0.851 | 0.168 | 29.87 | 0.907 | 0.247 |
| ✓ | | ✓ | 27.30 | 0.796 | 0.242 | 23.92 | 0.851 | 0.169 | 29.85 | 0.907 | 0.246 |
| | ✓ | ✓ | 27.30 | 0.796 | 0.243 | 23.87 | 0.852 | 0.167 | 30.02 | 0.908 | 0.247 |
| ✓ | ✓ | ✓ | **27.34** | **0.798** | **0.241** | **23.99** | **0.852** | **0.166** | **30.12** | **0.909** | **0.245** |

Table 4: Ablations on the core sampling strategy: Metropolis–Hastings (MH) vs. Stochastic Gradient Langevin Dynamics (SGLD). We denote the number of Gaussians next to the method.

| | Mip-NeRF360 | | | Tanks&Temples | | | Deep Blending | | |
|---|---|---|---|---|---|---|---|---|---|
| | PSNR ↑ | SSIM ↑ | LPIPS ↓ | PSNR ↑ | SSIM ↑ | LPIPS ↓ | PSNR ↑ | SSIM ↑ | LPIPS ↓ |
| 3DGS + MH (0.85M) | **26.87** | 0.782 | 0.260 | **23.66** | **0.843** | **0.173** | **29.46** | **0.898** | **0.258** |
| 3DGS + SGLD (0.85M) | 26.80 | **0.789** | **0.250** | 23.47 | 0.841 | 0.178 | 29.45 | **0.898** | 0.259 |

Table 5: Ablations on relocation and the multi-view importance score. Results presented for 3DGS use the same number of Gaussians as ours.

| | Mip-NeRF360 | | | Tanks&Temples | | | Deep Blending | | |
|---|---|---|---|---|---|---|---|---|---|
| | PSNR ↑ | SSIM↑ | LPIPS ↓ | PSNR ↑ | SSIM↑ | LPIPS ↓ | PSNR ↑ | SSIM↑ | LPIPS ↓ |
| 3DGS | 26.62 | 0.761 | 0.299 | 23.38 | 0.828 | 0.214 | 29.27 | 0.897 | 0.272 |
| Ours w/o Relocation | 26.94 | 0.778 | 0.267 | 23.69 | 0.844 | 0.184 | 29.57 | 0.901 | 0.267 |
| Ours w/o Importance Score | 27.31 | 0.800 | 0.234 | 23.79 | 0.851 | 0.167 | 29.97 | 0.906 | 0.245 |
| Ours | 27.34 | 0.798 | 0.241 | 23.99 | 0.852 | 0.166 | 30.12 | 0.909 | 0.245 |

**Contribution of Metropolis-Hastings Sampling.** To measure the exact contribution of our core Metropolis-Hastings (MH) sampling strategy, we conducted an ablation study comparing our method directly against the Stochastic Gradient Langevin Dynamics (SGLD) approach from 3DGS-MCMC [22]. We implemented only the core sampling algorithms from each method while maintaining matched Gaussian counts to ensure a fair comparison. The results shown in Tab. 4 demonstrate that MH sampling consistently outperforms SGLD across all benchmark datasets, confirming the effectiveness of our sampling strategy.

Furthermore, to understand the source of our method's effectiveness, we conducted an ablation study in Tab. 5 that isolates the contributions of the sampling process and the multi-view importance score. Our experiments demonstrate that the method retains strong performance even when the importance scores are removed. This indicates that the underlying sampling mechanism is the primary driver of our approach, while the multi-view importance scores act as an intelligent guidance mechanism.

## 6 Conclusion

We present an adaptive sampling framework that unifies densification and pruning for 3D Gaussian Splatting via a probabilistic Metropolis-Hastings scheme. The importance-weighted sampler automatically targets visually or geometrically critical regions while avoiding redundancy. We derive a new, concise probabilistic formulation that links proposals to scene importance, enabling faster convergence without sacrificing accuracy. Our experiments show that our method matches or surpasses state-of-the-art quality with fewer Gaussians, and our method's simplicity invites further advances in 3D scene representation.

**Limitations and Future Work.** Our Metropolis-Hastings sampler, though aware of low-density regions, still prioritizes high-density areas, leading to outdoor scenes with sparser data performing worse than indoor scenes. Moreover, using multi-view photometric errors when proposing points adds roughly 5 minutes of extra training time per scene. Crafting multi-scale representations or dual-stream foreground/background samplers, and introducing faster optimization to reduce training time, would be promising directions for future research.

# Acknowledgments

This work was supported by IITP grant (RS-2021-II211343: AI Graduate School Program at Seoul National Univ. (5%), RS-2023-00227993: Detailed 3D reconstruction for urban areas from unstructured images (35%), and RS-2025-02303703: Realworld multi-space fusion and 6DoF free-viewpoint immersive visualization for extended reality (60%)) funded by the Korea government (MSIT).

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

# A  Extended Derivation for Eq. 8

In Bayesian statistics, the posterior distribution of the parameters $\Theta$ given data $\mathcal{D}$ is defined as:

$$p(\Theta \mid \mathcal{D}) \;=\; \frac{p(\mathcal{D} \mid \Theta)\, p(\Theta)}{\underbrace{\int p(\mathcal{D} \mid \Theta)\, p(\Theta)\, d\Theta}_{Z}}, \tag{21}$$

where $p(\mathcal{D} \mid \Theta)$ is the likelihood, $p(\Theta)$ is the prior, and $Z$ is the marginal likelihood.

In Eq. 7, we introduced a negative log-posterior

$$\mathcal{E}(\Theta) = \mathcal{L}(\Theta) + \lambda_v \sum_{v \in \mathcal{V}} \ln(1 + c_\Theta(v)) \tag{22}$$

In Eq. 5, we found out that $\ln p(\mathcal{D} \mid \Theta) \approx -\mathcal{L}(\Theta)$. Combining these two, we can derive:

$$\mathcal{E}(\Theta) = -\ln p(\mathcal{D} \mid \Theta) - \ln p(\Theta) \tag{23}$$

If we exponentiate $-\mathcal{E}(\Theta)$:

$$e^{-\mathcal{E}(\Theta)} = p(\mathcal{D} \mid \Theta)p(\Theta), \tag{24}$$

we can derive the unnormalized posterior density. To turn this into a proper probability distribution, we normalize by $Z = \int e^{-\mathcal{E}(\Theta)}\, d\Theta$, yielding:

$$\pi(\Theta) \;=\; \frac{1}{Z}\, e^{-\mathcal{E}(\Theta)} \;=\; \frac{p(\mathcal{D} \mid \Theta)\, p(\Theta)}{Z} \;=\; p(\Theta \mid \mathcal{D}) \tag{25}$$

As $\pi(\Theta)$ matches the posterior distribution form in Eq. 21, we can define it as the posterior density.

# B  Extended Derivation for Eq. 19

Below is the derivation for Eq. 19. Recall Eq. 7, which defines the negative log-posterior, also known as the energy function:

$$\mathcal{E}(\Theta) = \mathcal{L}(\Theta) + \lambda_v \sum_{v \in \mathcal{V}} \ln(1 + c_\Theta(v)) \tag{26}$$

Thus $\Delta \mathcal{E} = \mathcal{E}(\Theta') - \mathcal{E}(\Theta)$ can be calcuated as:

$$\begin{aligned}
\Delta \mathcal{E} &= \mathcal{E}(\Theta') - \mathcal{E}(\Theta) \\
&= \mathcal{L}(\Theta') + \lambda_v \sum_{v \in \mathcal{V}} \ln(1 + c_{\Theta'}(v)) - \left(\mathcal{L}(\Theta) + \lambda_v \sum_{v \in \mathcal{V}} \ln(1 + c_\Theta(v))\right) \\
&= (\mathcal{L}(\Theta') - \mathcal{L}(\Theta)) + \lambda_v \sum_{v \in \mathcal{V}} [\ln(1 + c_{\Theta'}(v)) - \ln(1 + c_\Theta(v))]
\end{aligned}$$

As we considered the case of having added exactly one Gaussian into voxel $v'$, we can define $c_{\Theta'}(v)$, the number of Gaussians in $v'$ after the move, as:

$$c_{\Theta'}(v') = c_\Theta(v') + 1 \tag{27}$$

Moreover, the sum collapses to a single term for voxel $v'$. Thus, continuing our derivation:

$$\begin{aligned}
\Delta \mathcal{E} &= (\mathcal{L}(\Theta') - \mathcal{L}(\Theta)) + \lambda_v \sum_{v \in \mathcal{V}} [\ln(1 + c_\Theta(v') + 1) - \ln(1 + c_\Theta(v'))] \\
&= \Delta \mathcal{L} + \lambda_v \ln\left(\frac{1 + c_\Theta(v') + 1}{1 + c_\Theta(v')}\right) \\
&= \Delta \mathcal{L} - \ln\left(\frac{1 + c_\Theta(v')}{1 + c_\Theta(v') + 1}\right)^{\lambda_v}
\end{aligned}$$

Setting $\left(\frac{1+c_\Theta(v')}{1+c_\Theta(v')+1}\right)^{\lambda_v}$ as $D(v')$, we can derive:

$$\Delta \mathcal{E} = \Delta \mathcal{L} - \ln D(v') \tag{28}$$

Now set $x = \frac{1}{1 + c_\Theta(v')}$. We can use the first-order Taylor series expansion (also known as the first-order Maclaurin expansion) to approximate

$$\lambda_v \ln(1 + x) \approx \lambda_v x = \frac{\lambda_v}{1 + c_\Theta(v')} \tag{29}$$

We can further approximate as follows:

$$\ln D(v') = -\lambda_v \ln\left(1 + \frac{1}{1 + c_\Theta(v')}\right)$$
$$= -\lambda_v \ln(1 + x) \approx -\lambda_v x$$
$$D(v') \approx e^{-\lambda_v x} \tag{30}$$

Using the Maclaurin expansion, we can approximate:

$$e^{-a} = \frac{1}{e^a} = \frac{1}{1 + a + \frac{a^2}{2} + O(a^3)} \approx \frac{1}{1 + a} \tag{31}$$

Thus we can approximate $e^{-\lambda_v x}$ as $\frac{1}{1 + \lambda_v x}$

$$D(v') \approx e^{-\lambda_v x} \approx \frac{1}{1 + \frac{\lambda_v}{1 + c_\Theta(v')}} \approx \frac{1}{1 + \lambda_v c_\Theta(v')} \tag{32}$$

## C  Extended Derivation for Surrogate

This section explains why we approximate $-\Delta\mathcal{L}$ to $I(i)$.

Let's say we insert a tiny splat $\{\varepsilon\, g_i\}$ to the current state $\Theta$. We'll call this state $\Theta'$. By the first-order term in the multivariate Taylor expansion in the direction of $g_i$:

$$\Delta\mathcal{L} = \mathcal{L}(\Theta') - \mathcal{L}(\Theta) \approx \left\langle \nabla_{g_i} \mathcal{L}(\Theta),\, \varepsilon\, g_i \right\rangle \tag{33}$$

where $\nabla_{g_i} \mathcal{L}(\Theta)$ is the gradient of the loss w.r.t. the Gaussian's parameters and the inner product measures the first-order change when adding $\varepsilon\, g_i$.

Since $g_i$ only affects $p_i = \lfloor \Pi(\mathbf{x}_i) \rfloor$, we can show:

$$-\Delta\mathcal{L} \propto \alpha\, O(p_i) + \beta\, \mathrm{SSIM}_{\mathrm{agg}}(p_i) + \gamma\, \mathcal{L}1_{\mathrm{agg}}(p_i) = z_i \tag{34}$$

As stated in Eq. 12 and 13, we set:

$$I(i) = \sigma(z_i) = \frac{1}{1 + e^{-z_i}} \tag{35}$$

The Maclaurin series for $\sigma(z) = \frac{1}{1 + e^{-z}}$ around 0 is:

$$\sigma(z) = \sigma(0) + \sigma'(0)\, z + \frac{\sigma''(0)}{2!}\, z^2 + \frac{\sigma^{(3)}(0)}{3!}\, z^3 + \cdots$$
$$= \frac{1}{2} + \frac{1}{4}z - \frac{1}{48}z^3 + \cdots \approx \frac{1}{2} + \frac{1}{4}z$$

Thus, $\sigma(z_i) \approx \frac{1}{2} + \frac{1}{4}z_i$. Since our Metropolis-Hastings rule uses only relative magnitudes of $\sigma(z_i)$, the constant and the scale are absorbed, yielding $\sigma(z_i) \propto z_i$. Finally, if we put everything together:

$$-\Delta\mathcal{L} \approx z_i \approx \sigma(z_i) = I(i) \tag{36}$$

## D  Algorithms

Our algorithms for our coarse-fine Metropolis-Hastings sampling framework and overall pipeline are summarized in Alg. 1 and 2.

---

**Algorithm 1** Coarse–Fine Metropolis–Hastings Sampling for 3DGS

---

**Require:** Scene $\Theta = \{g_i\}$ where $g_i = (\mathbf{x}_i, \Sigma_i, c_i, \alpha_i)$
    View set $\mathcal{C}$; iteration $t$, total steps $t_{\max}$
    Voxel size $v$; density penalty $\lambda_v$
    Proposal stds $(\sigma_c, \sigma_f)$; batch sizes $(B_c, B_f)$
    Importance weights $(\alpha, \beta, \gamma)$; opacity threshold $\tau$

    **// Select diverse view subset for error aggregation**
1: $\mathcal{C}_t \leftarrow \text{VIEWSUBSET}(\mathcal{C}, t, t_{\max})$

    **// Compute multi-view error maps (SSIM, L1)**
2: $\text{SSIM}_{\text{agg}}, \mathcal{L}1_{\text{agg}} \leftarrow \frac{1}{|\mathcal{C}_t|} \sum_{c \in \mathcal{C}_t} \left[ 1 - \text{SSIM}(I^{(c)}, \hat{I}^{(c)}), \ |I^{(c)} - \hat{I}^{(c)}| \right]$

    **// Relocate Gaussians with low opacity**
3: **for** $g_i \in \Theta$ where $\alpha_i \leq \tau$ **do**
4:     $\text{RELOCATE}(g_i)$
5: **end for**

    **// Compute importance scores per Gaussian**
6: **for** $g_i \in \Theta$ **do**
7:     $p_i \leftarrow \Pi(\mathbf{x}_i)$                ▷ Project center to image plane
8:     $O(p_i) \leftarrow \alpha_i$             ▷ Use projected Gaussian opacity
9:     $S(i) \leftarrow \alpha\, O(p_i) + \beta\, \text{SSIM}_{\text{agg}}(p_i) + \gamma\, \mathcal{L}1_{\text{agg}}(p_i)$
10:     $I(i) \leftarrow \sigma(S(i))$
11: **end for**
12: $c(\cdot) \leftarrow \text{VOXELCOUNTS}(\Theta, v)$

    **// Metropolis–Hastings insertion (Coarse → Fine)**
13: **for** $(\sigma, B) \in \{(\sigma_c, B_c),\ (\sigma_f, B_f)\}$ **do**
14:     Sample $\mathcal{I}$ of size $B$ from $\Theta$ with $\Pr(i) \propto I(i)$
15:     **for** $i \in \mathcal{I}$ **do**
16:         $\mathbf{x}' \leftarrow \mathbf{x}_i + \mathcal{N}(0, \sigma^2 \mathbf{I})$
17:         $v' \leftarrow \text{VOXELINDEX}(\mathbf{x}', v)$
18:         $D(v') \leftarrow \frac{1}{1 + \lambda_v\, c(v')}$
19:         $\rho \leftarrow I(i) \cdot D(v')$            ▷ Acceptance probability
20:         **if** $\text{UNIFORM}(0, 1) < \rho$ **then**
21:             $\Theta \leftarrow \Theta \cup \{(\mathbf{x}', \Sigma_i, c_i, \alpha_i)\}$
22:         **end if**
23:     **end for**
24: **end for**

25: **return** $\Theta$

---

---

**Algorithm 2** Overall pipeline of Metropolis-Hastings 3DGS

---

1: **Input:** $M, S, C, A$ (SfM Points, Covariances, Colors, Opacities); image dimensions $w, h$.
2: **Output:** Optimized and densified attributes $M, S, C, A$.
3: $i \leftarrow 0$                              ▷ Iteration count
4: **while** not converged **do**
5:     $\{V, \hat{I}\} \leftarrow \text{SAMPLETRAININGVIEW}()$     ▷ Camera $V$ and image
6:     $I \leftarrow \text{RASTERIZE}(M, S, C, A, V)$
7:     $L \leftarrow \text{LOSS}(I, \hat{I})$
8:     $S_{\text{SSIM}, i} \leftarrow \text{SSIM}(I, \hat{I})$
9:     $S_{\text{L1}, i} \leftarrow \text{L1}(I, \hat{I})$
10:     $[M, S, C, A] \leftarrow \text{ADAM}(\nabla L)$           ▷ Update attributes
11:     **if** $\text{ISREFINEMENTITERATION}(i)$ **then**
12:         **for** Current Scene $\Theta$ **do**
13:             $\text{METROPOLIS\_HASTINGS\_SAMPLING}(M, S, C, A)$
14:         **end for**
15:     **end if**
16:     $i \leftarrow i + 1$
17: **end while**
18: **return** $M, S, C, A$

---

# E   Per-Scene Results

In this section, we provide the full per-scene evaluations. We evaluated our method, 3DGS [20] and 3DGS-MCMC [22] on various datasets and scenes. We provide the per-scene results for Mip-NeRF 360 [1], Tanks & Temples [23] and Deep Blending [17] datasets in Tab. A and B. All evaluations have been conducted on a NVIDIA GeForce RTX 3090.

Table A: Full per-scene results evaluated on Mip-NeRF 360 dataset.

| Type of Dataset | | | Indoor | | | Outdoor | | | | | |
|---|---|---|---|---|---|---|---|---|---|---|---|
| Scene | | Bonsai | Counter | Kitchen | Room | Bicycle | Flowers | Garden | Stump | Treehill | Average |
| 3DGS [20] | PSNR | 32.14 | 28.98 | 31.25 | 31.65 | 25.10 | 21.33 | 27.31 | 26.64 | 22.54 | 27.44 |
| | SSIM | 0.940 | 0.906 | 0.931 | 0.927 | 0.747 | 0.589 | 0.857 | 0.770 | 0.635 | 0.811 |
| | LPIPS | 0.206 | 0.202 | 0.117 | 0.197 | 0.244 | 0.359 | 0.122 | 0.216 | 0.347 | 0.223 |
| | Train (mm:ss) | 28:04 | 43:25 | 31:05 | 26:29 | 42:41 | 31:21 | 43:16 | 34:00 | 30:53 | 34:34 |
| | # of Gaussians | 1249672 | 1174274 | 1763905 | 1478813 | 5728256 | 3480107 | 5667876 | 4494105 | 3522801 | 3175479 |
| | Storage (MB) | 296.56 | 277.73 | 419.18 | 349.76 | 1354.80 | 823.09 | 1340.52 | 1062.91 | 833.18 | 750.86 |
| 3DGS-S [20] | PSNR | 31.37 | 28.84 | 30.91 | 31.04 | 23.55 | 20.10 | 26.27 | 24.84 | 22.64 | 26.62 |
| | SSIM | 0.935 | 0.909 | 0.924 | 0.918 | 0.616 | 0.488 | 0.794 | 0.677 | 0.590 | 0.761 |
| | LPIPS | 0.205 | 0.196 | 0.134 | 0.214 | 0.410 | 0.478 | 0.230 | 0.381 | 0.444 | 0.299 |
| | Train (mm:ss) | 26:42 | 28:04 | 33:59 | 25:22 | 25:54 | 23:58 | 28:17 | 23:02 | 28:00 | 27:02 |
| | # of Gaussians | 777782 | 698871 | 623992 | 764539 | 741748 | 695495 | 811695 | 697792 | 690517 | 722492 |
| | Storage (MB) | 183.96 | 165.29 | 147.58 | 181.02 | 175.43 | 164.49 | 191.98 | 165.04 | 163.32 | 170.90 |
| 3DGS-MCMC [22] | PSNR | 32.48 | 29.27 | 31.41 | 32.15 | 25.13 | 21.64 | 26.71 | 27.21 | 23.09 | 27.69 |
| | SSIM | 0.949 | 0.920 | 0.930 | 0.933 | 0.745 | 0.595 | 0.832 | 0.791 | 0.648 | 0.816 |
| | LPIPS | 0.176 | 0.177 | 0.126 | 0.186 | 0.275 | 0.377 | 0.170 | 0.247 | 0.370 | 0.234 |
| | Train (mm:ss) | 32:08 | 37:27 | 30:25 | 35:37 | 31:56 | 25:39 | 24:16 | 23:50 | 26:01 | 29:42 |
| | # of Gaussians | 777782 | 698871 | 623992 | 764539 | 741748 | 695495 | 811695 | 697792 | 690517 | 722492 |
| | Storage (MB) | 183.96 | 165.29 | 147.58 | 181.02 | 175.43 | 164.49 | 191.98 | 165.04 | 163.32 | 170.90 |
| Ours | PSNR | 32.52 | 29.30 | 31.35 | 32.15 | 24.53 | 20.99 | 26.49 | 25.95 | 22.80 | 27.34 |
| | SSIM | 0.950 | 0.918 | 0.930 | 0.934 | 0.711 | 0.556 | 0.819 | 0.743 | 0.621 | 0.798 |
| | LPIPS | 0.171 | 0.174 | 0.123 | 0.179 | 0.289 | 0.384 | 0.185 | 0.279 | 0.384 | 0.241 |
| | Train (mm:ss) | 41:05 | 46:59 | 41:20 | 46:31 | 34:58 | 35:13 | 31:36 | 32:46 | 37:05 | 38:37 |
| | # of Gaussians | 777782 | 698871 | 623992 | 764539 | 741748 | 695495 | 811695 | 697792 | 690517 | 722492 |
| | Storage (MB) | 183.96 | 165.29 | 147.58 | 181.02 | 175.43 | 164.49 | 191.98 | 165.04 | 163.32 | 170.90 |

Table B: Full per-scene results evaluated on Tanks & Temples and Deep Blending datasets.

| Dataset | | Tanks&Temples | | | Deep Blending | | |
|---|---|---|---|---|---|---|---|
| Scene | | Train | Truck | Average | Dr Johnson | Playroom | Average |
| 3DGS [20] | PSNR (↑) | 21.89 | 25.37 | 23.63 | 29.08 | 29.97 | 29.53 |
| | SSIM (↑) | 0.814 | 0.882 | 0.848 | 0.901 | 0.907 | 0.904 |
| | LPIPS (↓) | 0.207 | 0.147 | 0.177 | 0.244 | 0.244 | 0.244 |
| | Train (mm:ss) | 13:22 | 19:09 | 16:15 | 29:26 | 26:20 | 27:53 |
| | # of Gaussians | 1085904 | 2575516 | 1830710 | 3308886 | 2320688 | 2814787 |
| | Storage (MB) | 256.83 | 609.14 | 432.99 | 782.59 | 548.87 | 665.73 |
| 3DGS-S [20] | PSNR (↑) | 21.80 | 24.95 | 23.38 | 28.99 | 29.54 | 29.27 |
| | SSIM (↑) | 0.791 | 0.865 | 0.828 | 0.892 | 0.901 | 0.897 |
| | LPIPS (↓) | 0.245 | 0.183 | 0.214 | 0.280 | 0.263 | 0.272 |
| | Train (mm:ss) | 19:16 | 18:05 | 18:41 | 20:18 | 20:23 | 20:21 |
| | # of Gaussians | 788942 | 756104 | 772523 | 766469 | 735409 | 750939 |
| | Storage (MB) | 186.60 | 178.83 | 182.72 | 181.28 | 173.93 | 177.61 |
| 3DGS-MCMC [22] | PSNR (↑) | 22.58 | 26.02 | 24.30 | 29.37 | 30.30 | 29.84 |
| | SSIM (↑) | 0.834 | 0.889 | 0.862 | 0.902 | 0.910 | 0.906 |
| | LPIPS (↓) | 0.196 | 0.143 | 0.170 | 0.260 | 0.248 | 0.254 |
| | Train (mm:ss) | 15:50 | 15:45 | 15:48 | 26:28 | 23:39 | 25:03 |
| | # of Gaussians | 788942 | 756104 | 772523 | 766469 | 735409 | 750939 |
| | Storage (MB) | 186.60 | 178.83 | 182.72 | 181.28 | 173.93 | 177.61 |
| Ours | PSNR (↑) | 22.44 | 25.54 | 23.99 | 29.77 | 30.47 | 30.12 |
| | SSIM (↑) | 0.821 | 0.883 | 0.852 | 0.904 | 0.913 | 0.909 |
| | LPIPS (↓) | 0.198 | 0.134 | 0.166 | 0.249 | 0.240 | 0.245 |
| | Train (mm:ss) | 20:38 | 19:40 | 20:09 | 32:27 | 29:54 | 31:10 |
| | # of Gaussians | 788942 | 756104 | 772523 | 766469 | 735409 | 750939 |
| | Storage (MB) | 186.60 | 178.83 | 182.72 | 181.28 | 173.93 | 177.61 |

# F  Comparison of the Number of Gaussians

Table C contrasts the number of Gaussians needed by our approach and by 3DGS for each scene. In every case, our method attains equal or superior rendering quality while requiring substantially fewer Gaussians.

Table C: Comparison of our method and 3DGS on the number of Gaussian center points used to represent scenes in the Mip-NeRF 360 [1], Tanks & Temples [23], and Deep Blending [17] datasets.

| Dataset | Mip-NeRF 360 | | | | | | | | | Tanks&Temples | | Deep Blending | |
|---|---|---|---|---|---|---|---|---|---|---|---|---|---|
| Method | Bonsai | Counter | Kitchen | Room | Bicycle | Flowers | Garden | Stump | Treehill | Train | Truck | Dr Johnson | Playroom |
| 3DGS [20] | 1,249,672 | 1,174,274 | 1,763,905 | 1,297,736 | 4,888,552 | 2,887,138 | 5,667,876 | 4,324,049 | 3,224,118 | 1,085,904 | 2,580,408 | 3,308,886 | 2,320,688 |
| Ours | **777,782** | **698,871** | **623,992** | **765,370** | **741,748** | **695,495** | **811,695** | **697,792** | **690,517** | **788,942** | **756,104** | **766,469** | **735,409** |

# G  Broader Impacts

As our method focuses on the core and theoretical problem of 3D Scene Reconstruction, our work has the potential for positive societal impact on various downstream applications using 3D Gaussian Splatting [20]. As we reduce the reliance on heuristics and the number of Gaussians used in a scene, we expect our work to enhance many downstream applications that require less memory usage. However, as we did not introduce any fundamental new content generation capabilities through our work, there is little potential for our work to have a negative societal impact beyond the ethical considerations already present in the field of 3D reconstruction.

# H  Dataset Licenses

We used the following datasets:

- Mip-NeRF360 [1]: The license term is unavailable. Available at https://jonbarron.info/mipnerf360/

- Tanks and Temples [23]: Published under the Creative Commons Attribution 4.0 International (CC BY 4.0) License. Available at https://www.tanksandtemples.org/license/

- Deep Blending [17]: The license term is unavailable. Available at http://visual.cs.ucl.ac.uk/pubs/deepblending/

