# OpenReview forum: "Metropolis-Hastings Sampling for 3D Gaussian Reconstruction"
_NeurIPS.cc/2025/Conference — NeurIPS 2025 poster_

### Official Review · Reviewer_6AXt · 2025-06-04

**Clarity:** 2
**Significance:** 2
**Originality:** 3
**Rating:** 4
**Confidence:** 4

**Summary:**

This work proposes an improved Gaussian Splatting framework based on Metropolis-Hastings Sampling. Following the formulation and overall framework of 3DGS-MCMC, this work adopts Metropolis-Hastings Sampling to accept or reject newly added Gaussians. Besides, it considers the number of Guassians in each subvoxel (as a voxel-wise prior) to constraint the overall model size, and includes photometric error as long as opacity to construct Gaussian importance. Experimental results on several datasets show the proposed method is comparable with baseline algorithms.

**Questions:**

1. It would be better to provide results of Mip-NeRF360 and Tanks&Temples in Figure 1.
2. It would be better to provide models with varying number of Gaussians.
3. It would be better to provide ablation study to show that Metropolis-Hastings Sampling is better than Stochastic Gradient Langevin Dynamics in 3DGS-MCMC with a fair comparison (i.e., without the voxel-wise prior and photometric error as importance).

**Ethical Concerns:**

["NO or VERY MINOR ethics concerns only"]

**Final Justification:**

After reading the comments from other reviewers and the author response, Reviewer 6AXt acknowledges the technical contribution of MH sampling presented in this work. However, the limited performance improvement somewhat constrains the overall impact of the contribution. Therefore, the reviewer maintains the rating as a borderline accept.

**Limitations:**

yes

**Paper Formatting Concerns:**

There is no major formatting issue.

**Quality:**

3

**Strengths And Weaknesses:**

### Paper Strengths


It is reasonable and intuitive to incorporate the Metropolis-Hastings sampling method for the MCMC framework.




### Major Weaknesses

Abstract
1. Misleading clarification. This work claims that 'Traditional 3DGS methods heavily rely on heuristic based density-control mechanisms ... Our framework overcomes these limitations...' (line 3 to 8). Note that this limitation is dealed by 3DGS-MCMC with their state transition, and this work follows the overall pipeline of 3DGS-MCMC for this problem.

Related Work
1. Lack of related work for Sec. 2.3. Considering that Gaussians importance has been discussed and adopted by a great number of works, many studies on Gaussian densification have been overlooked. Besides, considering that this work highlights their contribution for efficient reconstruction (' a reduced number of Gaussians'), many studies on efficient Gaussian models are also ignored.

Preliminary
1. It would be better to include the introduction of 3DGS-MCMC and Metropolis-Hastings Sampling in this part.

Method
1. Confusing description (line 176). What is 'voxel' in the context of Guassian Splatting and how to construct voxels.
2. Cherry picking (Figure 1). In Figure 1., the proposed method shows better performance and converges faster than 3DGS-MCMC. However, according to Table 1, the proposed method only performs better than 3DGS-MCMC on deepblending. What about Mip-NeRF360 and Tanks&Temples?

Experiments
1. Limited improvement (Table 1). The proposed method shows limited improvements on most scans.
2. Performance with varying number of Gaussians. Although this work focuses on the case with limited number of Guassians, it would be better to provide models with varying number of Gaussians.
3. Ablation study. Considering that the key contribution of this work is to leverage Metropolis-Hastings Sampling for MCMC framework, it would be better to provide ablation study to show that Metropolis-Hastings Sampling is better than Stochastic Gradient Langevin Dynamics in 3DGS-MCMC with a fair comparison (i.e., without the voxel-wise prior and photometric error as importance).



### Minor Weaknesses

typo
1. line 30: samples->sampling

---

> ### Author Rebuttal · Authors · 2025-07-31
>
> We sincerely appreciate your insightful comments and questions. We have tried to respond to all of your concerns as follows. Moreover, thank you for pointing out the typo; we will revise it for our final manuscript.
>
> ## [W1] Misleading Clarification
>
> We acknowledge the reviewer's concern about our positioning. While we follow a similar overall pipeline (relocation and loss formulation) and address the same fundamental problem of replacing heuristic density control, we believe our characterization is accurate, as we propose a fundamentally different solution approach to the same problem. Specifically, 3DGS-MCMC employs SGLD (Stochastic Gradient Langevin Dynamics) that perturbs all Gaussian parameters with noise and accepts every move, whereas our method uses Metropolis-Hastings sampling that targets specific locations based on multi-view error aggregation and employs principled accept/reject decisions. We will revise our final manuscript to better acknowledge 3DGS-MCMC's pioneering role while more clearly stating how our MH-based approach with multi-view importance scoring represents a distinct methodological contribution to the same problem space. Our intent was not to diminish prior work, but rather to demonstrate an alternative probabilistic framework that achieves faster convergence.
>
> ## [W2] Lack of Related Work
>
> **Efficient Reconstruction**
> We have included the following works on efficient reconstruction in Sec. 2.2 of our paper [9, 22, 36, 38, 48], but we will further add additional more recent works on efficient reconstruction [e1, e2, e3] to Sec. 2.2. These works focus on deterministic geometric analysis or post-hoc uncertainty pruning on converged models. Unlike these methods, our probabilistic framework reduces the number of Gaussians through principled Bayesian sampling that dynamically adapts to scene-specific reconstruction signals.
>
> [e1] Papantonakis, Panagiotis, et al. "Reducing the memory footprint of 3d gaussian splatting." Proceedings of the ACM on Computer Graphics and Interactive Techniques 7.1 (2024): 1-1
>
> [e2] Hanson, Alex, et al. "Speedy-splat: Fast 3d gaussian splatting with sparse pixels and sparse primitives." Proceedings of the Computer Vision and Pattern Recognition Conference. 2025.
>
> [e3] Hanson, Alex, et al. "Pup 3d-gs: Principled uncertainty pruning for 3d gaussian splatting." Proceedings of the Computer Vision and Pattern Recognition Conference. 2025.
>
> **Densification**
> We have included related work regarding densification in Sec. 2.3 [4, 20, 30, 52], but as densification was a rapidly growing field of interest during the past year, we may have unintentionally overlooked recent studies. We will add recent densification works using neural prediction [d1] or deterministic geometric splitting [d2]. These methods can face limitations, such as generative approaches may not generalize well to novel scene types [d1], and geometric methods use fixed rules that may not adapt to diverse scene complexity [d2]. In contrast, our framework provides theoretical convergence guarantees via stochastic exploration of the solution space.
>
> [d1] Nam, Seungtae, et al. "Generative Densification: Learning to Densify Gaussians for High-Fidelity Generalizable 3D Reconstruction." Proceedings of the Computer Vision and Pattern Recognition Conference. 2025.
>
> [d2] Deng, Xiaobin, et al. "Efficient density control for 3d gaussian splatting." arXiv preprint arXiv:2411.10133 (2024).
>
> ## [W3] Preliminary
> We will move the introduction of 3DGS-MCMC and Metropolis-Hastings Sampling to the preliminary in the final manuscript.
>
> ## [W4] Confusing Description
> In the context of our paper, a voxel is the axis-aligned cube of edge length v, which is used purely for counting how many Gaussians occupy each bit of world space. To construct the voxel grid, we first lay a regular cubic lattice of size v across the scene’s coordinate frame. After that, the centre of every Gaussian is mapped to one of these voxels by dividing its 3D position by v and taking the integer part. We then record only how many Gaussians fall inside each voxel, $c_\Theta(v)$. These occupancy counts feed the sparsity voxel-wise prior as denoted in Eq. 4, which penalizes voxels that hold more than one or two Gaussians. Note that as we only use the voxels for counts, and never for rendering, we don’t need to store a full 3D grid.
>
> ## [W5/Q1] Results in Figure 1.
>
> As we are not allowed to provide images, we will provide the statistics regarding timing. Our method shows faster convergence for both Mip-NeRF360 and Tanks and Temples datasets, and shows faster convergence even when measured by the average of all the datasets. We provide the statistics of the convergence time when averaged across all 13 scenes below, and include the "98% of Final PSNR" as another reference.
>
> |                     | Ours     | 3DGS-MCMC |
> |---------------------|----------|-----------|
> | 95% of Final PSNR | 8.7 mins | 9.0mins   |
> | 98% of Final PSNR | 11.9 mins| 13.4mins  |
>
> We also provide a “time-to-equal-PSNR” metric, as suggested by reviewer U3jc, below.
>
> | Target PSNR  | Time - Ours           | Time - 3DGS-MCMC       |
> |--------------|----------------------|------------------------|
> | 21           | 16.30s / 0.27 mins   | 17.08s / 0.28 mins     |
> | 24           | 61.34s / 1.02 mins   | 98.38s / 1.64 mins     |
> | 27           | 287.01s / 4.78 mins  | 341.64s / 5.69 mins    |
> | 30           | 851.52s / 14.19 mins | 983.05s / 16.38 mins   |
>
> Our method consistently shows faster convergence compared to 3DGS-MCMC. We will update the figure to reflect all datasets in our final manuscript.
>
> ## [W6] Limited Improvement
> Our method shows mixed quantitative results on MipNeRF-360 and Tanks & Temples, which is a point we acknowledge in our paper's limitations section.
>
> Qualitatively, our method is on par with or even superior to baselines in capturing the fine details of the primary subjects, as can be seen in Fig. 2 and our supplementary video. However, as discussed in our limitations, the lower quantitative metrics are predominantly caused by our model's difficulty in representing vast, low-frequency backgrounds, such as the sky. Since these metrics are averaged over the entire image, imperfections in the large background regions can penalize the overall score.
>
> The high-quality results of reconstructing indoor scenes demonstrate our method's strong potential when scale variance is more manageable. On these scenes, our method achieves superior metrics across all measures, solidifying that the core algorithmic contributions are sound. This trade-off stems from the extreme scale variance (exceeding 170,000x) inherent to unbounded outdoor scenes, where our model struggles to simultaneously optimize for the compact, detailed Gaussians of the foreground and the massive, smooth Gaussians required for the background. As we state in our limitations, adapting our model to robustly handle both ends of this scale spectrum is a clear direction for future work, and this can be addressed by exploring techniques regarding multi-scale representation.
>
> ## [W7/Q2] Varying Number of Gaussians
>
> We provide results of our method, 3DGS-MCMC and 3DGS, with varying numbers of Gaussians across all scenes. Note that the performance of models is similar across varying numbers of Gaussians.
>
> **Average Number of Gaussians across all scenes = 0.53M**
>
> | Method | Mip-NeRF360 | Tanks&Temples | Deep Blending |
> |--------|-------------|---------------|---------------|
> |        | PSNR$\uparrow$/SSIM$\uparrow$/LPIPS$\downarrow$ | PSNR $\uparrow$/SSIM$\uparrow$/LPIPS$\downarrow$ | PSNR $\uparrow$/SSIM$\uparrow$/LPIPS$\downarrow$ |
> | MH-3DGS    | 27.14 / 0.790 / 0.253              | 23.75 / 0.847 / 0.179              | 29.99 / 0.906 / 0.251              |
> | 3DGS       | 26.46 / 0.756 / 0.301              | 23.30 / 0.824 / 0.221              | 29.27 / 0.897 / 0.277              |
> | 3DGS-MCMC  | 27.39 / 0.808 / 0.246              | 24.02 / 0.854 / 0.181              | 29.84 / 0.905 / 0.264              |
>
> **Average Number of Gaussians across all scenes = 1.40M**
>
> | Method | Mip-NeRF360 | Tanks&Temples | Deep Blending |
> |--------|-------------|---------------|---------------|
> |        | PSNR$\uparrow$/SSIM$\uparrow$/LPIPS$\downarrow$ | PSNR $\uparrow$/SSIM$\uparrow$/LPIPS$\downarrow$ | PSNR $\uparrow$/SSIM$\uparrow$/LPIPS$\downarrow$ |
> | MH-3DGS    | 27.58 / 0.810 / 0.212              | 24.06 / 0.857 / 0.153              | 30.04 / 0.911 / 0.233              |
> | 3DGS       | 27.08 / 0.786 / 0.261              | 23.52 / 0.842 / 0.192              | 29.60 / 0.903 / 0.255              |
> | 3DGS-MCMC  | 27.88 / 0.829 / 0.207              | 24.46 / 0.867 / 0.157              | 29.90 / 0.908 / 0.246              |
>
> ## [W8/Q3] Ablation Study
>
> Thank you for the valuable feedback. To directly compare Metropolis-Hastings (MH) Sampling with SGLD, we conducted an ablation study where both methods had only their key algorithm implemented. We removed our design components (voxel-wise prior, importance score) and also took out opacity and scaling regularizers from both methods. This setup creates the most direct comparison between our MH sampling and the SGLD approach.
>
> | Method | Mip-NeRF360 | Tanks&Temples | Deep Blending |
> |--------|-------------|---------------|---------------|
> |        | PSNR$\uparrow$/SSIM$\uparrow$/LPIPS$\downarrow$ | PSNR $\uparrow$/SSIM$\uparrow$/LPIPS$\downarrow$ | PSNR $\uparrow$/SSIM$\uparrow$/LPIPS$\downarrow$ |
> | 3DGS + MH (0.85M) | 26.87 / 0.782 / 0.260 | 23.66 / 0.843 / 0.173 | 29.46 / 0.898 / 0.258 |
> | 3DGS + SGLD (0.85M) | 26.80 / 0.789 / 0.250 | 23.47 / 0.841 / 0.178 | 29.45 / 0.898 / 0.259 |
>
> The results demonstrate that our Metropolis-Hastings sampling strategy provides consistent improvements over 3DGS-MCMC's SGLD approach. This result indicates that the MH sampling strategy provides a reliable advantage in reconstruction quality.

---

> > ### Comment · Reviewer_6AXt · 2025-08-05
> >
> > Thank you for the author response. After reading the comments from other reviewers and the author response, Reviewer 6AXt acknowledges the technical contribution of MH sampling presented in this work. However, the limited performance improvement somewhat constrains the overall impact of the contribution. Therefore, the reviewer maintains the rating as a borderline accept.

---

> > > ### Author Response · Authors · 2025-08-06
> > >
> > > We thank the reviewer for acknowledging the technical contributions of our work and for their constructive discussion. While we recognize that the performance improvements are modest, we believe our Metropolis-Hastings sampling approach's faster convergence compared to 3DGS-MCMC represents a meaningful step forward.

---

### Official Review · Reviewer_PUCv · 2025-07-01

**Clarity:** 3
**Significance:** 3
**Originality:** 3
**Rating:** 4
**Confidence:** 3

**Summary:**

The authors propose a novel approach to improve the 3DGS sampling strategy. They represent the importance of gaussians through measuring their multi-view pixel errors, and model the density control process in 3DGS based on Metropolis-Hastings sampling, such that densification process can be performed by generating new gaussian proposals with probability according to their importance. Subsequently, they can reduce the heuristic dependence in the density control process. Experiments show that the new approach can outperform vanilla 3DGS with fewer numbers of gaussians, and achieve faster convergence comparing to 3DGS-MCMC.

**Questions:**

My main concern is about the advantages comparing to 3DGS-MCMC, could the authors illustrate more on the significance of the proposed method comparing to 3DGS-MCMC? And it would be helpful to provide some failure cases on the Mip-NeRF 360 and TNT datasets and the corresponding insights to help future research, since in Figure 2 the proposed method successfully reconstruct the distant details comparing to 3DGS-MCMC, while underperforming by 0.3dB PSNR in Table 1.

**Ethical Concerns:**

["NO or VERY MINOR ethics concerns only"]

**Limitations:**

Please see the weaknesses and questions above.

**Paper Formatting Concerns:**

N.A.

**Quality:**

3

**Strengths And Weaknesses:**

Strengths:
1. Novelty: Bringing the Metropolis-Hastings sampling process into 3DGS density control is novel and reasonable, which models the probability of densification through measuring the multi-view pixel errors to focus on the areas with larger errors, while avoiding introducing strongly heuristic strategies.
2. Soundness: The paper provides strict mathematical derivation of applying the Metropolis-Hastings sampling to 3DGS density control.

Weaknesses:
1. Performance: The proposed method seems to slightly underperform comparing to 3DGS-MCMC after convergence. The major significance of the proposed method comparing to 3DGS-MCMC is a bit unclear.

---

> ### Author Rebuttal · Authors · 2025-07-31
>
> We sincerely appreciate your insightful comments and questions. We have tried to respond to all of your concerns as follows.
>
> ## [W1/Q1] Performance - Significance of MH and Failure Cases
>
> In order to directly compare our methodology with 3DGS-MCMC, we conducted an ablation study where both methods had only their key algorithm implemented. We removed our design components (voxel-wise prior, importance score) and also took out opacity and scaling regularizers from both methods. We believe that this setup creates the most direct comparison between our MH sampling and the SGLD approach.
>
> | Method | Mip-NeRF360 | Tanks&Temples | Deep Blending |
> |--------|-------------|---------------|---------------|
> |        | PSNR$\uparrow$/SSIM$\uparrow$/LPIPS$\downarrow$ | PSNR $\uparrow$/SSIM$\uparrow$/LPIPS$\downarrow$ | PSNR $\uparrow$/SSIM$\uparrow$/LPIPS$\downarrow$ |
> | 3DGS + MH (0.85M) | 26.87 / 0.782 / 0.260 | 23.66 / 0.843 / 0.173 | 29.46 / 0.898 / 0.258 |
> | 3DGS + SGLD (0.85M) | 26.80 / 0.789 / 0.250 | 23.47 / 0.841 / 0.178 | 29.45 / 0.898 / 0.259 |
>
> The results demonstrate that our Metropolis-Hastings sampling strategy itself provides consistent improvements over 3DGS-MCMC's SGLD approach. Furthermore, this result indicates that the MH sampling strategy provides a reliable advantage in reconstruction quality.
>
> Moreover, beyond the significance highlighted in our paper, key significances of our MH framework are as follows:
>
> 1. Automatic scene-complexity adaptation. As mentioned in the paper, 3DGS-MCMC must commit to a fixed budget of Gaussians before training, which can under-fit simple scenes or waste memory on complex ones. By contrast, our accept/reject chain keeps inserting points only while the posterior still yields photometric gain, so the optimization automatically stops at the scene-appropriate count with no prior complexity.
>
> 2. Dimension-selective updates that accelerate convergence. 3DGS-MCMC injects Langevin noise into every parameter of every Gaussian each step, which inflates stochastic gradient variance and forces smaller learning rates. On the other hand, we restrict stochastic moves to positions only. This trims the update dimensionality and lets each accepted proposal deliver a larger effective step. Thus, the PSNR curve for our method is smoother and achieves convergence earlier than 3DGS-MCMC (Fig. 1), confirming that the dimensionality cut translates into faster practical convergence.
>
> However, as mentioned in our paper, in sparse outdoor scenes such as Flowers, Stump, etc. (Mip-NeRF 360) or Train, Truck (T&T), our PSNR trails that of 3DGS-MCMC. These gaps arise not from missed high-frequency details, which we capture well as can be seen in Fig. 2 and the supplementary video, but from faint background strata that carry little image gradient yet contribute to pixel-wise error. Our MH sampler weighs proposals by multi-view photometric discrepancy; once high-error foreground patches are saturated, the residual background error is dispersed across many low-density voxels, so their individual acceptance probabilities fall below the threshold, leaving some regions undersampled. To improve these failure cases, adopting adaptive temperature annealing to MH acceptance or adopting multi-scale representations or dual-stream foreground/background samplers could be viable solutions, and open grounds for future work.

---

> > ### Author Response · Authors · 2025-08-06
> >
> > Dear Reviewer PUCv,
> >
> > Thank you so much for your time and effort in reviewing our work and providing insightful feedback. As the reviewer-author discussion window is closing soon, we would greatly appreciate it if you could review our rebuttal at your earliest convenience.
> >
> > We welcome any further discussions and value the opportunity for continued improvement of our work.
> >
> > Best regards,
> >
> > Paper 9768 Authors

---

> > ### Comment · Reviewer_PUCv · 2025-08-07
> >
> > Thank you for your detailed reponse, which has solved my questions. I will keep my rating.

---

> > > ### Author Response · Authors · 2025-08-07
> > >
> > > We sincerely thank the reviewer for the response, we are glad that your questions have been well addressed. Thank you once again for your constructive feedback.

---

> ### Comment · Area_Chair_8Rdk · 2025-08-07
> **Please engage to the discussions!**
>
> Dear Reviewer,
>
> I would like to invite you to the discussions with the authors. At least, please carefully read the others' reviews and authors' responses, and mention if the rebuttals addressed the concerns or not.
>
> To facilitate discussions, the Author-Reviewer discussion phase has been extended by 48h till Aug 8, 11:59 pm AoE; but to have enough time to exchange opinions, please respond as quickly as possible.
>
> Thanks,
>
> Your AC

---

### Official Review · Reviewer_gybH · 2025-07-03

**Clarity:** 4
**Significance:** 3
**Originality:** 3
**Rating:** 4
**Confidence:** 3

**Summary:**

This paper targets the stage of densification and sparsification in 3D gaussian splitting (3DGS). In the 3DGS work this stage is conducted with a heuristic rule based on opacity and photometric loss gradients on each Gaussian primitive. This paper shows that the process can be viewed as a sampling process, where a new configuration of Gaussian sets are sampled and accepted/rejected. The authors propose to use Metroplis Hasting as the sampling framework, with proposal generation composed of Gaussian densification and relocation. An method to calculate importance scores is introduced to conduct the sampling.

The authors tested the proposed method on common 3DGS benchmarks. The method can lead to on-par reconstruction quality with vanilla 3DGS but with much fewer number of Gaussians, saving compute and storage. Authors also compare the method with 3DGS-MCMC, another work formulating the specific stage as sampling problem. The propose method yields faster convergence than this baseline.

**Questions:**

One interesting question I have is whether the method's effectiveness comes from the sampling process or the multi-view importance score. The original 3DGS uses gradients in a single-view batch as a cue for the densification, while in this work, a large set of views are used to evaluate whether a Gaussian needs to be "cloned." It would be great if the authors could provide insight on this matter.

**Ethical Concerns:**

["NO or VERY MINOR ethics concerns only"]

**Final Justification:**

The authors have addressed my questions regarding this work. The merit of this work is as described in the reviews. I would like to keep my rating as borderline accept.

**Limitations:**

Yes.

**Quality:**

3

**Strengths And Weaknesses:**

+ The formulation of densification and sparsification as point configuration sampling provides a systematic view of the originally heuristics based stage which is empirically found to be critical to 3DGS reconstruction quality.
+ The proposed important scored based on multi-view reconstruction error seems to provide a better cue for densification and sparsification over the original 3DGS's loss gradients which is dependent on mini-batch sampling.
+ In experiment, the method yield both fast convergence and smaller number of Gaussian than the baselines, showing its effectiveness.
+ The authors have made good effort to make the theoretical part of this work easy to understand.

Weakness
- As the authors also pointed out, the method requires evaluating multiple views for the importance score in each sampling step, which adds up to the total training cost.
- In measuring the per Gaussian importance score, a simple surrogate is used to reduce complexity. Not much discussion is provided on what is the quality impact of this choice.

---

> ### Author Rebuttal · Authors · 2025-07-31
>
> We sincerely appreciate your insightful comments and questions. We have tried to respond to all of your concerns as follows.
>
> ## [W1] Multiple-Views for Importance Score - Training Cost
>
> While our method does require multi-view evaluation for importance scoring, hence leading to an increase in the total training cost, the faster convergence shown in Fig. 1 demonstrates that this overhead is worthwhile in practice. Our method reaches 95% of the final PSNR faster than 3DGS-MCMC in both iteration count and wall-clock time, meaning the improved sampling efficiency more than compensates for the additional per-iteration cost. The key insight is that spending slightly more time per iteration to make better sampling decisions results in needing fewer total iterations to converge, and this is achieved even with our current unoptimized implementation.
>
> Additionally, our approach uses only a camera subset (annealed from broad to focused coverage) rather than the full training set, which helps control the computational burden. Moreover, our method achieves comparable or better reconstruction quality while using dramatically fewer Gaussians (as shown in Table 4), leading to significant memory savings. Nevertheless, we hope to explore more efficient importance score computation or adaptive view selection strategies to further reduce this overhead.
>
> ## [W2] Discussion on the surrogate
>
> Computing the exact loss reduction ($-\Delta\mathcal L$) for each candidate Gaussian would require re-rendering the entire scene from all viewpoints in the selected camera subset for every proposed splat. This would increase computational complexity from O(N) to O(N×K×M), where N is the number of candidates, K is the number of cameras, and M is the scene complexity, making the method computationally impractical. To solve this problem, we derived a surrogate by analyzing the first-order multivariate Taylor expansion based on inserting a small Gaussian in the scene. (Mentioned in our paper in Appendix C)
>
> Indeed, the best way to measure the quality impact would be a direct comparison, but a direct empirical comparison between the surrogate and exact $-\Delta\mathcal L$ would require computing full multi-view rendering errors for thousands of candidate Gaussians, which was precisely the computational burden our surrogate is designed to avoid. This creates a paradox where validating the surrogate's accuracy would negate its computational benefits.
>
> Nevertheless, we believe that the quality impact of selecting a surrogate to calculate $-\Delta\mathcal L$ is minimal, as the surrogate maintains the relative ordering of candidate importance. Since our surrogate is derived from a rigorous first-order approximation, and the logistic function $\sigma(z)$ is strictly monotonic, the relative ordering is mathematically preserved. For example, if candidate A truly provides greater improvement than candidate B, it would be $I(A) > I(B)$ in our surrogate. This ordering preservation is sufficient for quality maintenance because the MH acceptance test ensures that better candidates receive higher acceptance probabilities, guiding the sampler toward high-impact regions. Over many iterations, this statistical bias toward high-importance areas drives convergence to optimal Gaussian distributions, as evidenced by our results showing maintained or improved reconstruction quality.
>
> ## [Q1] Model's effectiveness coming from sampling or importance score
>
> We conducted an ablation study by removing the multi-view importance score and other components in our method. We also reported the average number of Gaussians across all scenes in the parentheses.
>
> | Method | Mip-NeRF360 | Tanks&Temples | Deep Blending |
> |--------|-------------|---------------|---------------|
> |        | PSNR$\uparrow$/SSIM$\uparrow$/LPIPS$\downarrow$ | PSNR $\uparrow$/SSIM$\uparrow$/LPIPS$\downarrow$ | PSNR $\uparrow$/SSIM$\uparrow$/LPIPS$\downarrow$ |
> | No Importance Score (0.75M) | 27.31 / 0.800 / 0.234 | 23.79 / 0.851 / 0.167 | 29.97 / 0.906 / 0.245 |
> | MH-3DGS (0.75M) | 27.34 / 0.798 / 0.241 | 23.99 / 0.852 / 0.166 | 30.12 / 0.909 / 0.245 |
>
> Based on the study, our method without the multi-view importance scores still achieves strong performance, outperforming 3DGS-MCMC on all three metrics on Deep Blending and LPIPS in Tanks & Temples. This shows that replacing heuristic-based densification with principled Metropolis-Hastings sampling is effective.
>
> To further clarify, our multi-view component is strategically integrated into the sampling framework, not as the primary driver of performance, but as an intelligent guidance mechanism that makes the probabilistic sampling more informed and consistent. Overall, our sampling methodology demonstrates that principled sampling with robust guidance is effective.

---

> > ### Comment · Reviewer_gybH · 2025-08-05
> >
> > Thanks for the detailed response. I agree with the argument that evaluating the full importance score is cost-prohibitive. But maybe some visualization on a subset of the Gaussian is still feasible and could strengthen the point of order preservation from an empirical perspective. This does not hurt the quality of this work. I am keeping my recommendation to accept.

---

> > > ### Author Response · Authors · 2025-08-06
> > >
> > > We thank the reviewer for their recommendation to accept and for the constructive discussion. We appreciate the suggestion to add visualizations on a subset of Gaussians and will make sure to incorporate this into the final manuscript to strengthen our point.

---

### Official Review · Reviewer_U3jc · 2025-07-07

**Clarity:** 3
**Significance:** 1
**Originality:** 2
**Rating:** 4
**Confidence:** 4

**Summary:**

This paper re-casts the splitting and pruning operations of 3DGS training into the framework of Metropolis-Hastings (MH) Monte Carlo methods, similar in spirit to the recent MCMC-3DGS paper (Both are Markov Chain methods, except that MH adds in an accept probability for proposals). The method also introduces a number of heuristics, such as a voxel occupancy prior to discourage "overpopulation" of solved areas, a per-pixel importance field based on a combination of photometric error and accuracy, and new heuristics for "coarse" and "fine" gaussian proposals. Performance metrics are provided on benchmark datasets, which do not consistently outperform MCMC-3DGS. A limited series of ablations justify some of their design choices.

**Questions:**

How much _exactly_ does MH contribute to performance _gains_ in a direct "apples-to-apples" comparison between your method and 3DGS-MCMC. I realize a direct comparison may not be strictly possible, but it doesn't appear as though every pains were taken to make this comparison possible. Furthermore, how much do all your design heuristics advance reconstruction metrics, if at all? A strong indication that MH improves performance in all cases would shift me to "accept" or a thorough analysis of why MH alone fails.

What is the point of aggregating the same pixel from different images in Eq. 9? presumably the same pixel in different images represent rays passing through entirely different parts of the 3D scene, so It's unclear how this could be useful information. Presumably you are computing the importance per-Gaussian, then averaging views, so it seems as though the notation is wrong?

**Ethical Concerns:**

["NO or VERY MINOR ethics concerns only"]

**Final Justification:**

The authors have provided evidence that their method converges faster than 3DGS-MCMC. I really would have preferred to see a graph for each dataset, as there is concievable ambiguity in how these numbers were arrived at (average all scenes training PSNR first, then timing, or each scene individually, then average). Regardless, I'm trusting the authors that there are genuine performance gains, and am revising my reccommendation to "borderline accept".

**Limitations:**

Yes.

**Paper Formatting Concerns:**

None noticed.

**Quality:**

3

**Strengths And Weaknesses:**

The paper is weak in it's quantitative metrics, just barely producing a few SotA results on the Deep Blending datasets, but is only two scenes. On the larger set of scenes in MipNerf-360 and Tanks and Temples, 3DGS-MCMC still achieves superior metrics. It's also difficult to draw conclusions from the qualitative results - the inset images do show superior quality with their method, but without access to the high-res images or error map visualizations, there could concievably be worse areas in their images as well.

Any performance gains in timings are also unclear. Figure 1 overlays wall-time vertical lines on an iteration axis, which are entirely different measures. This would seem to imply that the relative performance of each method is equivalent, but is not directly discussed?. I'm also not sure what the point of a "95% of final PSNR" metric is - it almost seems like metric only constructed demonstrate the method in a good light. Why is the more typical "time to equal PSNR" metric not appropriate in this setting?

At a high level, I really want to like this paper, as I'm a fan of more mathematically justifiable techniques such as Metropolis-Hastings. However, computer vision is as much "art" as it is "science", and this paper does not _fully_ disentangle the gains made by heuristics (Eq. 4, 10, 12, 13, round-robin scheduling of sec 4.2.1), and those from the addition of MH. MH could very well be a consistent win, but this paper does not make that clear, making it of limited interest to the NeurIPS audience, hence my reccommendation of reject.

---

> ### Author Rebuttal · Authors · 2025-07-31
>
> We sincerely appreciate your insightful comments and questions. We have tried to respond to all of your concerns as follows.
>
> ## [W1] Weakness in quantitative results
>
> We would like to first highlight that our core Metropolis-Hastings (MH) sampling strategy demonstrates improvements over 3DGS-MCMC's SGLD approach in a direct comparison (please see our detailed analysis in Q1/W3). However, we acknowledge that our overall method shows mixed quantitative results on MipNeRF-360 and Tanks & Temples, which is a point we also acknowledge in our paper's limitations section.
>
> Qualitatively, our method is on par with or even superior to baselines in capturing the intricate geometry and fine details of the primary subjects, as can be seen in Fig. 2. However, the low quantitative metrics are predominantly caused by our model's difficulty in representing vast, low-frequency backgrounds, such as the sky. Since these metrics are averaged over the entire images, background imperfections can disproportionately penalize scores despite the high-quality foreground reconstruction.
>
> The results on the indoor scenes with manageable scale variance demonstrate our method's strong potential. Our method outperforms the baselines in these scenes, suggesting that core algorithmic contributions are sound. This challenge stems from the extreme scale variance (exceeding 170,000x) inherent to unbounded outdoor scenes, where our model struggles to simultaneously optimize for the compact, detailed foreground Gaussians and the massive, smooth background Gaussians. Adapting our model to robustly handle both ends of this scale spectrum is a clear direction for future work, and this can be addressed by exploring techniques such as multi-scale representation.
>
> ## [W2] Clarity on performance gains on timing
> In Fig. 1, the horizontal axis indeed counts training iterations. We imposed vertical markers that correspond to actual wall-clock time at which each method reaches a chosen quality threshold. We intended to give a single, compact view of how fast along the iteration trajectory each method would be experienced in real time.
>
> Regarding the reviewer's concern with the "95% of final PSNR" metric, we would like to clarify that we were not intending to demonstrate our method in a good light; we were rather unaware of the typical “time to equal PSNR” metric. We initially chose “95% of the final PSNR” as the metric, as a confidence-interval-like view of near-final quality that is directly comparable across methods, especially as absolute PSNR ceilings differ across scenes. Moreover, using 95% is adopted as a practical convergence metric in recent vision works; for instance, [1] uses 95% to compare optimization speed.
>
> Following the reviewer's suggestion, we provide the “time to equal PSNR” metrics averaged across all scenes for both our method and 3DGS-MCMC. As the approximate highest average PSNR value was 30 dB, we selected our final target PSNR as 30 dB. The results show that our method converges approximately 2.18 minutes faster than 3DGS-MCMC. We also show lower target PSNR values to show that our method consistently converges faster. We will include these metrics in our final manuscript.
>
> Note that we used the official implementation provided by 3DGS-MCMC and conducted experiments on NVIDIA GeForce RTX 3090 for both methods to ensure a fair comparison.
>
> | Target PSNR  | Time - Ours           | Time - 3DGS-MCMC       |
> |--------------|----------------------|------------------------|
> | 21           | 16.30s / 0.27 mins   | 17.08s / 0.28 mins     |
> | 24           | 61.34s / 1.02 mins   | 98.38s / 1.64 mins     |
> | 27           | 287.01s / 4.78 mins  | 341.64s / 5.69 mins    |
> | 30           | 851.52s / 14.19 mins | 983.05s / 16.38 mins   |
>
> [1] Zhang, Yunxiang, et al. "Image-gs: Content-adaptive image representation via 2d gaussians." *arXiv preprint arXiv:2407.01866* (2024)
>
> ## [Q1/W3] Direct Comparison of MH & 3DGS-MCMC and impact of design heuristics
>
> Thank you for this important question. The concern about disentangling the gains from our core Metropolis-Hastings (MH) framework versus our other design choices is crucial. To address this directly, we have conducted experiments to measure the contribution of each component.
>
> **1. Quantifying the Contribution of Metropolis-Hastings (MH) in an "Apples-to-Apples" Comparison with 3DGS-MCMC**
>
> To isolate the exact contribution of our core MH sampling strategy, we ran experiments with both our method and 3DGS-MCMC, both methods with only their key algorithmic methodology implemented. We believe that this setup creates the most direct "apples-to-apples" comparison between our Metropolis-Hastings (MH) proposal strategy and 3DGS-MCMC’s Stochastic Gradient Langevin Dynamics (SGLD) approach.
>
> | Method | Mip-NeRF360 | Tanks&Temples | Deep Blending |
> |--------|-------------|---------------|---------------|
> |        | PSNR$\uparrow$/SSIM$\uparrow$/LPIPS$\downarrow$ | PSNR $\uparrow$/SSIM$\uparrow$/LPIPS$\downarrow$ | PSNR $\uparrow$/SSIM$\uparrow$/LPIPS$\downarrow$ |
> | 3DGS + MH (0.85M) | 26.87 / 0.782 / 0.260 | 23.66 / 0.843 / 0.173 | 29.46 / 0.898 / 0.258 |
> | 3DGS + SGLD (0.85M) | 26.80 / 0.789 / 0.250 | 23.47 / 0.841 / 0.178 | 29.45 / 0.898 / 0.259 |
>
> The results demonstrate that our Metropolis-Hastings sampling strategy provides consistent improvements over 3DGS-MCMC's SGLD approach. This result indicates that the MH sampling strategy provides a reliable advantage in reconstruction quality.
>
> **2. Disentangling Design Heuristics**
>
> To address the reviewers' concern regarding this, we wish to clarify that our model’s components are theoretically grounded designs that function as interdependent parts of our Metropolis-Hastings framework. Our paper theoretically derived how these components must work together. As mentioned in Eq. 18, the first term $\sigma\bigl(I(i)\bigr)$ represents the photometric gain of adding a new Gaussian, which is calculated from Eq. 10 and the second term $D(v’)$ represents the sparsity penalty, calculated from Eq. 4. A proposal is accepted only if the product of these two terms is high, and this ensures that our method prioritizes Gaussians that are photometrically useful and spatially efficient.
>
> To further disentangle our design components, we present a detailed ablation study breaking down our method. We start with pure MH Sampling, which is the MH sampling algorithm with the regularizers. We incrementally added our key design components, and all the experiments were conducted in the same setting as our original experiments, with no separate parameter tuning for individual components. The results are summarized in the table below.
>
> | Method | Mip-NeRF360 | Tanks&Temples | Deep Blending |
> |--------|-------------|---------------|---------------|
> |        | PSNR$\uparrow$/SSIM$\uparrow$/LPIPS$\downarrow$ | PSNR $\uparrow$/SSIM$\uparrow$/LPIPS$\downarrow$ | PSNR $\uparrow$/SSIM$\uparrow$/LPIPS$\downarrow$ |
> | MH (0.85M) | 27.42 / 0.797 / 0.237 | 23.87 / 0.852 / 0.167 | 29.96 / 0.907 / 0.245 |
> | MH + Eq. 4 (0.77M) | 27.31 / 0.800 / 0.234 | 23.79 / 0.851 / 0.166 | 29.97 / 0.906 / 0.243 |
> | MH + Eq. 10 (0.85M) | 27.37 / 0.797 / 0.236 | 23.81 / 0.853 / 0.163 | 29.94 / 0.905 / 0.245 |
> | MH + Eq.4, Eq.10 (1.45M) | 27.52 / 0.809 / 0.214 | 23.94 / 0.858 / 0.148 | 29.98 / 0.907 / 0.233 |
> | MH + Eq. 12, Eq. 13 (1.45M) | 27.53 / 0.807 / 0.223 | 23.97 / 0.853 / 0.164 | 29.99 / 0.903 / 0.243 |
> | MH-3DGS (0.75M) | 27.34 / 0.798 / 0.241 | 23.99 / 0.852 / 0.166 | 30.12 / 0.909 / 0.245 |
>
> The results show that naively combining our design components without the full, synergistic framework (e.g., Pure MH + Eq.4, Eq.10) leads to an inefficient model. While it achieves a high PSNR, it does so by nearly doubling the number of Gaussians to 1.45M. This demonstrates that the components, when not properly balanced by the complete acceptance rule, can lead to uncontrolled growth, chasing marginal gains at a significant computational cost.
>
> While Pure MH achieves a marginally higher PSNR on the Mip-NeRF360 dataset, our final MH-3DGS model achieves this competitive result using 12% fewer Gaussians (0.75M vs. 0.85M). With all the design components together, we are able to create a much more compact and robust performance across all scenes.
>
> ## [Q2] Equation 9
>
> To clarify the notation, when we refer to pixel p in Eq. 9, we mean the pixel coordinate where a specific Gaussian projects in each camera's image plane. Since the same 3D Gaussian will project to different pixel coordinates in different camera views, the aggregation is performed per-Gaussian across its projected pixel locations in multiple views, not at the same pixel coordinate across different images.
>
> The process works as follows: we compute residuals locally for each camera $c$ at the pixel location where each Gaussian projects, then average these residuals across the $k_t$ cameras in $C_t$. This aggregation pools error statistics across multiple perspectives of the scene for stable error estimates. The per-Gaussian importance is then evaluated based on where each Gaussian projects in the aggregated error maps.
>
> We agree with the reviewer that the current notation can be misleading, and we will revise the notation for the final manuscript. Thank you for pointing out this important issue.

---

> > ### Author Response · Authors · 2025-08-06
> >
> > Dear Reviewer U3jc,
> >
> > Thank you so much for your time and effort in reviewing our work and providing insightful feedback. As the reviewer-author discussion window is closing soon, we would greatly appreciate it if you could review our rebuttal at your earliest convenience.
> >
> > We welcome any further discussions and value the opportunity for continued improvement of our work.
> >
> > Best regards,
> >
> > Paper 9768 Authors

---

> ### Comment · Area_Chair_8Rdk · 2025-08-07
> **Please engage to the discussions!**
>
> Dear Reviewer,
>
> I would like to invite you to the discussions with the authors. At least, please carefully read the others' reviews and authors' responses, and mention if the rebuttals addressed the concerns or not.
>
> To facilitate discussions, the Author-Reviewer discussion phase has been extended by 48h till Aug 8, 11:59 pm AoE; but to have enough time to exchange opinions, please respond as quickly as possible.
>
> Thanks,
>
> Your AC

---

### Comment · Area_Chair_8Rdk · 2025-08-01
**Author-Reviewer Discussion Period (July 31 - Aug 6)**

The author rebuttals are now posted.

To reviewers:
Please carefully read the *all* reviews and author responses, and engage in an open exchange with the authors.
Please post the response to the authors as soon as possible, so that we can have enough time for back-and-forth discussion with the authors.

---

> ### Comment · Area_Chair_8Rdk · 2025-08-05
> **Discussion Period Ends Soon (Aug 6)!**
>
> Dear reviewers,
> Thanks so much for reviewing the paper. The discussion period ends soon. To ensure enough time to discuss this with the authors, please actively engage in the discussions with them if you have not done so.

---

### Decision · Program_Chairs · 2025-09-17

**Decision:**

Accept (poster)

**Comment:**

This paper proposes a densification and sparsification process in 3D Gaussian splatting (3DGS). The proposed method uses importance values of each Gaussian and uses Metropolis-Hastings sampling to achieve probability-based densification. Experiments show that the proposed method outperforms 3DGS with fewer numbers of gaussians, and achieves faster convergence compared to 3DGS-MCMC.

The reviewers valued the novelty of the paper for the new sampling process for 3DGS. Since the original 3DGS relies on the heuristics of the densification-sparsification processes, the sampling method would be beneficial. As reviewers mentioned, the 3DGS-MCMC aims at a similar goal, while the proposed method using Metropolis-Hastings sampling achieves faster convergence.

Overall, the reviewers' assessment weakly converges to the acceptance of the paper. Meanwhile, the AC suggests clarifying and emphasizing the technical difference and merit of using the proposed method over 3DGS-MCMC in the final version.